# Viral and immune dynamics of genital human papillomavirus infections in young women with high temporal resolution

Nicolas Tessandier[1,2]*, Baptiste Elie[1,2]*, Vanina Boué[2], Christian Selinger[2,3], Massilva Rahmoun[2], Claire Bernat[2,4], Sophie Grasset[2], Soraya Groc[2,5], Anne-Sophie Bedin[5], Thomas Beneteau[2], Marine Bonneau[6], Christelle Graf[6], Nathalie Jacobs[7], Tsukushi Kamiya[1], Marion Kerioui[8], Julie Lajoie[9], Imène Melki[1], Jean-Luc Prétet[10,11], Bastien Reyné[2], Géraldine Schlecht-Louf[12], Mircea T. Sofonea[5,13], Olivier Supplisson[1,14], Chris Wymant[15], Vincent Foulongne[5], Jérémie Guedj[8], Christophe Hirtz[16], Marie-Christine Picot[17], Jacques Reynes[18], Vincent Tribout[19], Édouard Tuaillon[5], Tim Waterboer[20], Michel Segondy[5], Ignacio G. Bravo[2], Nathalie Boulle[5], Carmen Lía Murall[2,21], Samuel Alizon[1,2]*

1 CIRB, CNRS, INSERM, Collège de France, Université PSL, Paris, France, 2 MIVEGEC, CNRS, IRD, Université de Montpellier, France, 3 Swiss Tropical and Public Health Institute, Basel, Switzerland, 4 CNRS UMR 5203, Institut de Génomique Fonctionnelle, Montpellier, France, 5 PCCEI, Univ. Montpellier, Inserm, EFS, Montpellier, France, 6 Department of Obstetrics and Gynaecology, Centre Hospitalier Universitaire de Montpellier, Montpellier, France, 7 Laboratory of Cellular and Molecular Immunology, GIGA Institute, University of Liège, Liège, Belgium, 8 IAME, INSERM, Université de Paris, Paris, France, 9 Department of Medical Microbiology, University of Manitoba, Winnipeg, Canada, 10 Université de Franche-Comté, CNRS, Chrono-environnement, Besançon, France, 11 Centre National de Référence Papillomavirus, CHRU de Besançon, France, 12 UMR996, Inflammation, Chemokines and Immunopathology, INSERM, Université Paris-Saclay, Orsay, France, 13 CHU de Nîmes, Nîmes, France, 14 Sorbonne Université, France, 15 Big Data Institute, Li Ka Shing Centre for Health Information and Discovery, Nuffield Department of Medicine, University of Oxford, Oxford, United Kingdom, 16 RMB-PPC, INM, Univ Montpellier, CHU Montpellier, INSERM CNRS, Montpellier, France, 17 Department of Medical Information (DIM), Centre Hospitalier Universitaire de Montpellier, Montpellier, France, 18 Department of Infectious and Tropical Diseases, Centre Hospitalier Universitaire de Montpellier, Montpellier, France, 19 Center for Free Information, Screening and Diagnosis (CeGIDD), Centre Hospitalier Universitaire de Montpellier, Montpellier, France, 20 German Cancer Research Center (DKFZ), Infections and Cancer Epidemiology, Heidelberg, Germany, 21 National Microbiology Laboratory (NML), Public Health Agency of Canada (PHAC), Canada

☯ These authors contributed equally to this work.
* nicolas.tessandier@ird.fr (NT); baptiste.elie@ens-paris-scaly.fr (BE); samuel.alizon@cnrs.fr (SA)

**Data Availability Statement:** The raw data and R scripts used to perform the analyses and generate the figures and table are available at the CNRS

## Abstract

Human papillomavirus (HPV) infections drive one in 20 new cancer cases, exerting a particularly high burden on women. Most anogenital HPV infections are cleared in less than two years, but the underlying mechanisms that favour persistence in around 10% of women remain largely unknown. Notwithstanding, it is precisely this information that is crucial for improving treatment, screening, and vaccination strategies. To understand viral and immune dynamics in non-persisting HPV infections, we set up an observational longitudinal cohort study with frequent on-site visits for biological sample collection. We enrolled 189 women aged from 18 to 25 and living in the area of Montpellier (France) between 2016 and 2020. We performed 974 on-site visits for a total of 1,619 months of follow-up. We collected data on virus load, local immune cell populations, local concentrations of cytokines, and circulating antibody titres. Using hierarchical Bayesian statistical modelling to simultaneously

Research Data repository at https://doi.org/10.57745/KJG0YZ.

**Funding:** This EVOLPROOF project has received funding from the European Research Council (ERC) under the European Union's Horizon 2020 research and innovation programme (grant agreement No 648963, to SA). The authors acknowledge further support from the Centre National de la Recherche Scientifique, the Institut de Recherche pour le Développement, the Fédération Hospitalière Universitaire InCH of Montpellier (to SA), the Fondation pour la Recherche Medicale (to TK), the Ligue contre le Cancer (to TB), the Agence Nationale de la Recherche contre le Sida (ANRS-MIE, to NT (grant # 21290), IM and OS (grant # 22485)) and to the Labex MemoLife (to BE). The funders had no role in study design, data collection and analysis, decision to publish, or preparation of the manuscript.

**Competing interests:** TW serves on advisory boards for MSD (Merck Sharp and Dohme). JR reports personal fees from Gilead (consulting and payment or honoraria for lectures, presentations, speaker's bureaus, manuscript writing, or educational events), Janssen (payment or honoraria for lectures, presentations, speaker's bureaus, manuscript writing, or educational events), Merck (payment or honoraria for lectures, presentations, speaker's bureaus, manuscript writing, or educational events), Theratechnologies (payment or honoraria for lectures, presentations, speaker's bureaus, manuscript writing, or educational events), and ViiV Healthcare (consulting and payment or honoraria for lectures, presentations, speaker's bureaus, manuscript writing, or educational events) and support for attending meetings and/or travel from Gilead and Pfizer, outside of the submitted work. All the other authors do not report any conflict of interest.

**Abbreviations:** AUC, area under the curve; BMI, body mass index; CI, confidence interval; CrI, credibility interval; Ct, cycle threshold; CyTOF, cytometry by time of flight; FC, fold change; FCM, flow cytometry; FDR, false discovery rate; GST, glutathione s-transferase; HPV, human papillomavirus; MSD, Meso Scale Discovery; PBS, phosphate-buffered saline; qPCR, quantitative polymerase chain reaction; RBC, red blood cells; RPMI, Roswell Park Memorial Institute; STI, sexually transmitted infection; UMAP, uniform manifold approximation and projection.

analyse the data from 164 HPV infections from 76 participants, we show that in two months after infection, HPV viral load in non-persisting infections reaches a plateau that lasts on average for 13 to 20 months (95% credibility interval) and is then followed by a rapid clearance phase. This first description of the dynamics of HPV infections comes with the identification of immune correlates associated with infection clearance, especially gamma-delta T cells and CXCL10 concentration. A limitation of this study on HPV kinetics is that many infection follow-ups are censored. Furthermore, some immune cell populations are difficult to label because cervical immunity is less well characterised than systemic immunity. These results open new perspectives for understanding the frontier between acute and chronic infections, and for controlling HPV-associated diseases, as well as for research on human cancers of infectious origin.

**Trial Registration:** This trial was registered is registered at ClinicalTrials.gov under the ID NCT02946346. This study has been approved by the Comité de Protection des Personnes (CPP) Sud Méditerranée I (reference number 2016-A00712-49); by the Comité Consultatif sur le Traitement de l'Information en matière de Recherche dans le domaine de la Santé (reference number 16.504); by the Commission Nationale Informatique et Libertés (reference number MMS/ABD/ AR1612278, decision number DR-2016–488), by the Agence Nationale de Sécurité du Médicament et des Produits de Santé (reference 20160072000007).

## Author summary

### Why was this study done?

- Despite human papillomaviruses (HPVs) being the most oncogenic viruses, we know little about non-persistent infections in young adults.

- Nearly one out of five women of age 25 has an HPV genital infection, but more than 90% of these clear in less than two years.

- The determinants of HPV genital infection clearance are largely unknown.

- In 2016, a longitudinal study was initiated to monitor variations in HPV virus load and the associated immune response.

### What did the researchers do and find?

- HPV virus load exhibits a rapid increase, a long plateau, and a steep decrease.

- Infections are estimated to have a median duration between 13 and 20 months.

- Immune patterns are associated with HPV infection, especially gamma-delta T cells and CXCL10 chemokine.

- Variations in virus load are primarily associated with the HPV genotype and variations in infection duration with host differences.

**What do these findings mean?**

- HPV infection clearance is associated with the activation of the immune response.

- Variations in virus load can help to optimise screening policies.

- Details of the immune response can help identify biomarkers and targets for immunotherapies.

## Introduction

Chronic infections by oncogenic human papillomaviruses (HPVs) cause nearly all cervical cancers, most anogenital cancers, and a substantial fraction of oropharyngeal cancers [1]. With 83% of HPV-induced cancers being cervical, women are the most affected by the 630,000 HPV-induced cancers reported worldwide in 2012 [1]. This burden, further exacerbated by millions of cases of anogenital warts caused by other HPVs [2], stems from the fact that these viruses cause among the most prevalent sexually transmitted infections (STIs), with a high transmission risk per sexual contact [3]. Fortunately, more than 90% of these infections do not persist for more than two years in young adults [4–7]. The factors driving infection clearance are poorly known and could involve the adaptive and/or the innate immune response [8], but also random events occurring during cell division [9,10]. Over the last two decades, safe and efficient vaccines have been developed that target the most oncogenic HPV genotypes (especially HPV16 and HPV18), as well as genotypes causing genital warts (HPV6 and HPV11) [11,12]. Notwithstanding, chronic infections by HPVs will remain a major public health issue for at least one generation because of low vaccine coverage in many countries and the decreased vaccine efficacy if administered after exposure [13].

Although often asymptomatic and benign, non-persisting (or "acute") HPV infections raise important challenges [14]. Foremostly, the quality of screening policies relies on the description of the natural history of the infection [15]. Understanding interactions between HPVs and the immune system may also shed new light on the factors that lead to clearance or chronicity, with implications for human cancers of infectious origin [16] and the development of immunotherapies [17]. Finally, non-persisting HPV infections represent a major reservoir of virus diversity, which could fuel an evolutionary response to vaccine-driven selective pressures [18].

The prevalence and duration of non-persisting HPV infections have been monitored for decades [4,5,19], yet we still know little about the immune response they might elicit and temporal variations in virus loads. Even from a cross-sectional perspective, HPV status has been associated with the presence of some cytokines [20–22] but insights about the cellular immune response remain limited. One practical reason that explains this matter of affairs is that HPV infections in young adults are usually asymptomatic. HPV vaccines randomised control trials enrolled participants prospectively [7,23,24] and their control arms greatly improved our knowledge of HPV natural history. However, they remained limited in two ways. First, the follow-up of the participants took place at least every six months, which is non-optimal given what we now know about the duration of these infections. Second, analysing the local cellular immunity raises technical challenges because cervical smears are fragile tissues that should be processed fresh for flow cytometry analyses [25]. Even more problematically, these samples contain a majority of keratinocytes, which are large, adhesive, and auto-fluorescent cells.

Therefore, identifying and quantifying leukocytes by flow cytometry in cervical smears constitutes a challenge rarely attempted in the context of HPV infections (but see [26] for an exception). Many insights about the cellular immune response to HPVs have been obtained from biopsies, which have more material and are less fragile than smears [27]. However, collecting such samples longitudinally from healthy women is ethically problematic. Furthermore, compared to a smear, a biopsy only samples a small region of the cervical area, which restricts their use to symptomatic HPV infections, e.g., with pre-cancerous lesions.

Investigating the dynamics of the virus genome load and the associated immune response during asymptomatic HPV infections required a dedicated longitudinal study interfaced with timely immunological analyses. To this end, we implemented the PAPCLEAR cohort study in Montpellier (France) during which 189 women aged from 18 to 25 years old were followed every two months until HPV infection clearance or for a maximum duration of 24 months [28]. The main characteristics of the cohort are summarised in Table 1. At each of the 974 on-site visits, biological samples were collected and participants filled in detailed sociodemographic, health, and behaviour questionnaires.

Additional issues had to be handled besides setting up the cohort itself. First, the flow cytometry data analysis proved too challenging to be performed with manual gating because of the number of samples to process simultaneously and because of the amount of noise in the data. This was addressed by implementing an unsupervised clustering pipeline. Second, the

**Table 1. PAPCLEAR cohort profile stratified by HPV infection status.** Significant differences according to a *t* test with a 5% significance threshold (unadjusted *p*-values) are shown in bold font. Participants who were positive at least once in the follow-up for a DEIA test (see the Methods) were labelled as "HPV positive," whereas participants who were never positive during the follow-up were labelled as "HPV negative." For time-varying variables such as BMI, stress level, and all variables identified with "last 2 weeks," we only report the value at the inclusion visit.

| | HPV negative | HPV positive | *p*-value |
|---|---|---|---|
| Number of participants (*n*) | 63 | 126 | |
| **Lifetime number of partners (mean (SD))** | 8.68 (8.41) | 12.06 (11.07) | 0.034 |
| **Vaccinated against HPV = Yes (%)** | 40 (63.5) | 57 (45.2) | 0.027 |
| Age at first visit (mean (SD)) | 21.43 (2.01) | 21.60 (2.02) | 0.576 |
| Age at menarchy (mean (SD)) | 12.49 (1.23) | 12.87 (1.39) | 0.067 |
| First intercourse (age) (mean (SD)) | 16.62 (2.14) | 16.37 (1.85) | 0.401 |
| **Duration of follow up (days) (mean (SD))** | 198.30 (188.62) | 292.09 (213.02) | 0.003 |
| BMI (mean (SD)) | 21.99 (2.82) | 22.48 (3.62) | 0.352 |
| Antibiotics (last 2 weeks) = Yes (%) | 2 (3.2) | 10 (7.9) | 0.343 |
| Menses (last 2 weeks) = Yes (%) | 35 (55.6) | 64 (50.8) | 0.643 |
| Smoking (%) | | | 0.299 |
| No | 39 (61.9) | 82 (65.1) | |
| Occasionally | 12 (19.0) | 14 (11.1) | |
| Regularly | 12 (19.0) | 30 (23.8) | |
| Lubricant use (last 2 weeks) = Yes (%) | 12 (19.0) | 18 (14.3) | 0.526 |
| Intercourse with regular partner (last 2 weeks) = Yes (%) | 40 (63.5) | 70 (55.6) | 0.375 |
| Intercourse with occasional partner (last 2 weeks) = Yes (%) | 7 (11.1) | 18 (14.3) | 0.704 |
| Stress level (%) | | | 0.861 |
| 0 (Min) | 13 (20.6) | 20 (15.9) | |
| 1 | 25 (39.7) | 54 (42.9) | |
| 2 | 18 (28.6) | 39 (31.0) | |
| 3 (Max) | 7 (11.1) | 13 (10.3) | |

BMI, body mass index; HPV, human papillomavirus.

virus load data analysis was complicated by the absence of prior knowledge. Indeed, earlier studies only considered HPV presence or absence, had few visits per participant, or had a long delay between two observations (and sometimes all three at once). However, at least one mechanistic within-host mathematical model investigated the course of HPV infections [29]. Even though it makes some strong assumptions, e.g., that infection clearance is solely driven by a population of immune effectors, it proved useful to narrow down the field of possibilities and guide our statistical analysis. In particular, one of its outcomes is that the stage-structured nature of the virus' life cycle, which is imposed by that of its resources, the epithelial cells, strongly shapes the dynamics of the infection. This manifests itself by the existence of a virus load plateau that can last several months before clearance and is typically absent in non-structured virus kinetics models [30]. In this manuscript, following the convention, we call "viral load" the genome viral load, i.e., the number of viral genome copies measured by qPCR, normalised by the quantity of a human gene copy number representing the amount of cells in the sample.

Thanks to the high temporal resolution of our clinical study, the richness of the biological data generated, and the quality of the statistical analyses, we generate new insights about the course of HPV infections and the associated immune response.

## Results

### Characterising the local immune response to HPV infections

HPV infections are notoriously poorly immunogenic, as only 40% to 60% of the women with positive results for HPV DNA at the cervix seroconvert to the incident viral genotype [31]. Since we did not expect to detect a strong signal at the systemic level, we focused our efforts on the local cellular immune response, using flow cytometry (FCM) to analyse cervical smears (see the Methods). As mentioned above, this was challenging because keratinocytes are highly adhesive and autofluorescent cells, thereby rendering the interpretation of the fluorescent signal from our markers for key cell surface proteins difficult. Building on an existing protocol [25] and software packages [32], we devised a pipeline to identify clusters of immune cell populations from the fluorescence signal associated with each of the marked cells through a non-supervised approach.

We first used this pipeline to identify leukocytes (CD45$^+$ cells) by simultaneously analysing data from 362 cervical smears. We then applied it to the resulting cell population to automatically delineate 20 cellular clusters, which we manually grouped into 11 distinct immune cell populations based on morphological and lineage markers. Given the lack of prior knowledge about the mucosal immune response to HPV infections, we used a broad panel of fluorescent antibodies to mark the cells. As a consequence, our ability to label some of the immune cell populations we identify (or "clusters") is limited. In the following, to facilitate the reading, we refer to our best interpretation. Additional details about these populations and their identification can be found in Figs A–C and Table B in S1 Supplementary Materials.

The most frequent leukocytes by far were CD16+ granulocytes (cluster I, 59.9% of the CD45$^+$ cells). We also identified three distinct populations of TCRγδ$^+$ cells (clusters III, VII, and IX), representing respectively 3.88%, 12.44%, and 0.73% of the CD45$^+$ cells (Fig A in S1 Supplementary Materials), as well as CD4 and CD8 T cells (respectively 1.15% and 0.78%). Other cell populations could not be assigned formally but display features of local antigen-presenting cells (cluster V) or NK cells (cluster II) (Figs A–C in S1 Supplementary Materials).

To analyse associations between the cervical cellular immune response and HPV status, we stratified the samples as negative or positive for a "focal" infection. To define the latter, we require that the same HPV genotype is detected at least during two consecutive visits (Fig O

and Table A in S1 Supplementary Materials). This distinction was made to avoid a spurious focus on "singletons," i.e., when an HPV genotype is detected at a single visit, because these events, sometimes referred to as "transient infections" [33], are a poor marker of actual infections [34]. The resulting restricted data set comprised 145 HPV–negative and 186 HPV-focal flow cytometry samples. To visualise the 11 immune cell populations and the composition differences between our two types of samples, we performed a uniform manifold approximation and projection (UMAP) clustering. As shown in Fig 1A, the increase in one of the TCRγδ populations (cluster VII in orange) is particularly pronounced. A differential abundance analysis (Fig 1B) confirmed that focal HPV–positive samples exhibited a lower proportion of CD4$^+$ T-cells (cluster X, in pink) than HPV–negative samples (fold change, FC, of 0.62, and adjusted $p$-value, p-adj, of 0.01) and a higher abundance of one of the TCRγδ populations (cluster VII, in orange) and of a population more challenging to formally identify (CD45$^{low}$ CD3$^-$ CD16$^-$, cluster VIII) (with respective FC of 1.71 with a p-adj of 0.0009, and of 1.41 with a p-adj of 0.021, Fig 1B and 1C, and Table C in S1 Supplementary Materials). NK cells, monocytes, and other leukocytes (clusters IV, V, and VI) were also rarer in focal HPV–positive samples (FCs of 0.63, 0.71, 0.67, with p-adj of 0.004, 0.01, and 0.021, respectively). These differences were statistically significant in a generalised linear mixed model including a random effect accounting for the multiple sampling per participant (Fig 1C).

To check whether anti-HPV vaccination status affected the local immune response, we performed a differential abundance model similar to the one shown in Fig 1B using anti-HPV vaccination status as the response variable. Out of the 11 cell populations identified, only the one consistent with monocytes (CD45$^+$ CD4int leukocytes, cluster V) was positively more abundant in vaccinated samples (FC of 1.44 with p-adj of 0.02, Table D in S1 Supplementary Materials). This suggests that the shift in immune cell composition we identify in focal HPV infection is unlikely to be explained by the vaccination status.

Differential expression analysis highlighted moderate changes regarding the fluorescence levels of our two activation markers, CD69 and CD161. Focusing on CD4 T cells, we could identify a reduced median signal intensity for CD69 in HPV–positive (focal) compared to HPV–negative samples, which suggests lower activation of these cells during the infection (cluster X, FC of 0.90 and p-adj of 0.04, Table E in S1 Supplementary Materials).

Building on earlier results from the literature [20–22], we quantified the concentrations of five cytokines in cervical secretions using Meso Scale Discovery (MSD) technology and normalised the values over total protein concentration. Associations with HPV status showed a tendency for lower titers of IL-17A in samples positive for an HPV focal infection (Fig 2A and Tables F and G in S1 Supplementary Materials).

To improve the characterisation of immune cells in samples positive for an HPV focal infection, we explored correlations between the proportion of each immune cell population in a cervical sample and the mucosal concentration of each cytokine (Fig 2B). Our linear models identified positive correlations between the concentration of IFNγ and the frequency of monocytes (cluster V), with a regression coefficient $\beta = 0.14$ and a $p$-value of 0.03 (see the Methods for details). IFNγ was also correlated with the frequency of CD4 and CD8 T cells (respectively cluster X and XI, with regression coefficients of $\beta = 0.13$ and $\beta = 0.11$ and $p$-values of 0.12 and 0.12). Finally, the frequency of monocytes (cluster V) was slightly correlated with CXCL10 concentration ($\beta = 0.09$, $p = 0.074$). We also identified a negative correlation between CXCL10 and what appear to be NK cells (cluster II, $\beta = -0.19$ and $p = 0.013$). A similar correlation analysis was run for HPV–negative samples (Fig D in S1 Supplementary Materials). We did not find any association between IFNγ and monocytes or T cells in these samples. Conversely, we found a clear negative association between some TCRγδ cells (cluster VII) and CXCL10 ($\beta = -0.25$, $p = 0.028$) in HPV–negative sample.

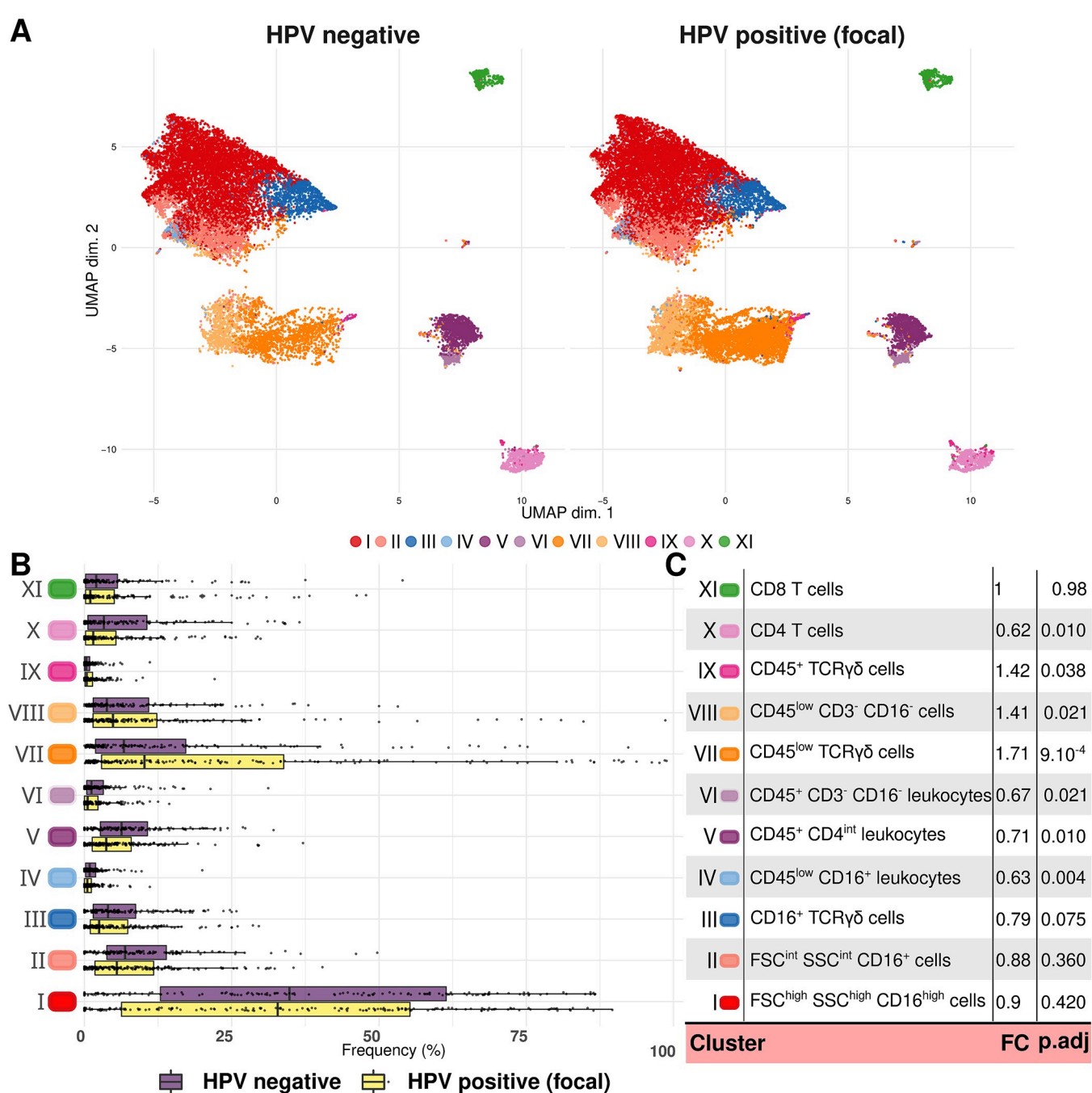

**Fig 1. Unsupervised clustering of flow cytometry data from cervical smears.** (A) Highlighting 11 homogeneous populations using UMAP clustering on data from samples without HPV or positive for a "focal" HPV infection. (B) Comparison of the clusters frequencies based on infection status, and (C) cluster annotation and FC values shown in panels A and B. FC were calculated by an abundance analysis with the diffcyt-DA-edgeR method adjusted with a Benjamini–Hochberg test. In panel B, each dot represents a single sample. Raw differential abundance results can be found in Table C in S1 Supplementary Materials. The code to generate this figure can be found in https://doi.org/10.57745/KJGOYZ. FC, fold change; HPV, human papillomavirus; UMAP, uniform manifold approximation and projection.

Our results on local cell population profiles are consistent with those on cytokines or chemokines production in the cervical area. For example, we find that IFNγ seems to be associated with monocytes, CD4, and CD8 T cells, which are known to produce it. Furthermore, the

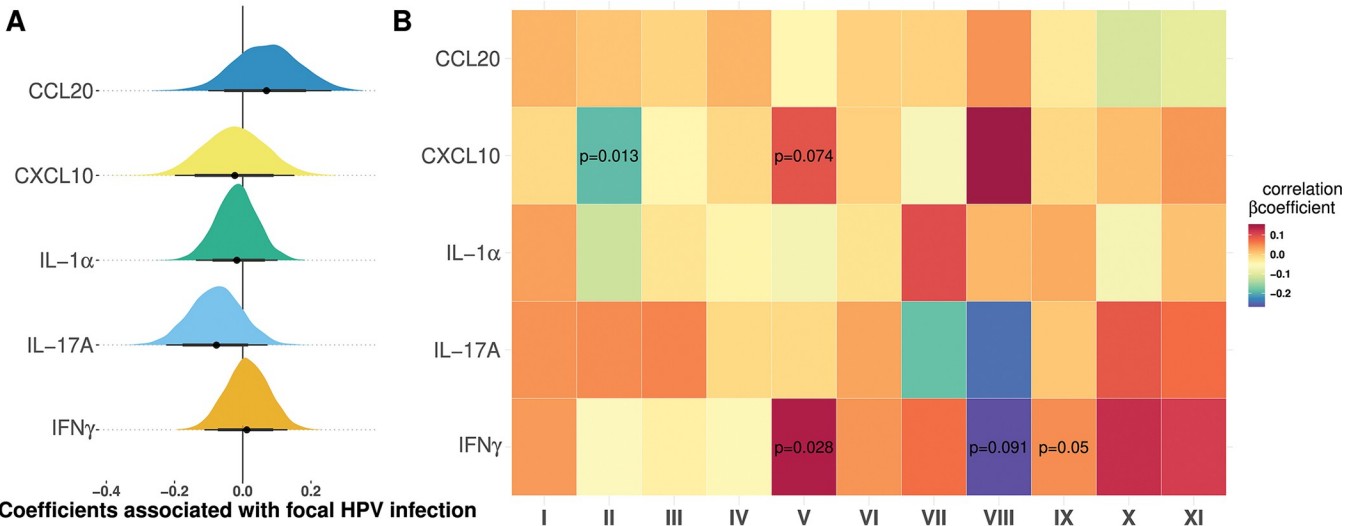

**Fig 2. Local immune response associated with HPV infections.** (A) Association between HPV focal status and five cytokines. We show the posterior distributions of a Bayesian model in which the log10 of each cytokine or chemokine is a predictor of the HPV status (focal or uninfected), assuming a correlated random effect for each participant (see Tables F and S7 in S1 Supplementary Materials for details). Thick lines and shaded areas show the 80% and 95% credibility intervals respectively. (B) For the HPV focal infections only, correlation matrix between the local density of five cytokines and the proportion of the 11 cell clusters from Fig 2. β represents the regression coefficient of linear regression with each cell population as the response variable (see Methods). For example, a β of 0.14 can be interpreted as follows: "a 10 percentage point increase in the CD8 T cells frequency is associated with a 1.4 percentage point increase in the IFNγ concentration." A similar correlation matrix for HPV–negative samples can be found in Fig D in S1 Supplementary Materials. The code to generate this figure can be found in https://doi.org/10.57745/KJGOYZ. HPV, human papillomavirus.

CXCL10 concentration seems to be negatively correlated with the size of the NK cell population (cluster II). NK cells are known to be recruited by CXCL10, so this negative correlation could be explained by differences in temporal dynamics or uncertainties in manual cluster assignment. Interestingly, most associations between cell clusters and cytokines production were specific to the HPV infection status.

## Viral load kinetics

In our cohort, there were 76 participants for whom we collected at least three clinical samples and detected at least one HPV genotype during their follow-up. For all of them, we estimated the virus load in the cervical smears collected at each on-site visit for 13 genotypes using a sensitive and specific quantitative polymerase chain reaction (qPCR) protocol [35]. The number of viral genome copies was normalised by twice that of a cellular gene (albumin). Overall, we could monitor 164 infections, including 66 left-censored infections where participants were enrolled as positive (40%), 34 right-censored infections where clearance was not observed (21%), 37 doubly censored infections (23%), and 27 complete infections (16%) (Fig 3 and Fig E in S1 Supplementary Materials). A total of 41 participants (54%) were coinfected, i.e., infected by more than one genotype simultaneously, during the follow-up.

Given the longitudinal nature of our data, we turned to the field of viral kinetics for our analysis [30]. As explained in the introduction, we did not find earlier clinical data to inform us about variations in HPV virus load over the course of an infection. However, we had two prior pieces of information to elaborate a statistical model. The first one was the mechanistic within-host model that concluded that the stage-structured nature of the infection led to a potentially long plateau phase before viral clearance [29]. The second one was the raw data itself. As shown by the data points in Fig 3, few of the follow-ups appear consistent with "classical" acute infection dynamics that have only a growth and a decrease phase. Furthermore,

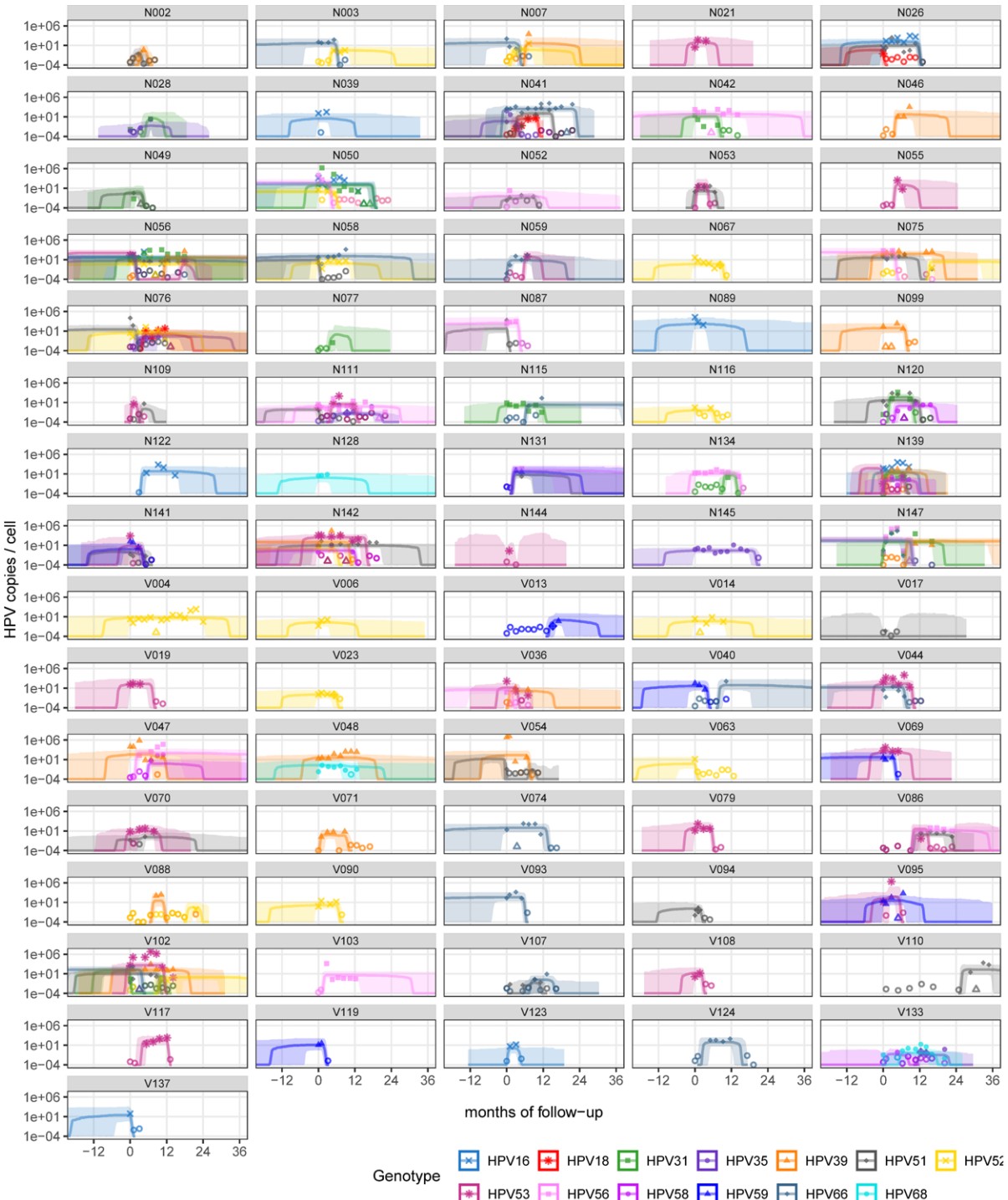

**Fig 3. Modelling virus load kinetics for 158 HPV genital infections in 76 women.** The dots indicate the data points and, therefore, the window of observation for each participant. Each panel corresponds to one participant and shows the number of HPV genome copies per number of human genome copies resulting from a three-slopes hierarchical Bayesian model. The lines show the trajectory of the posterior median value and shaded area the 95% CrI. Open circles indicate values below the limit of detection, open triangles indicate samples where the virus was detected but the viral load could not be estimated (see the Methods). Letters before the anonymity number (above each panel) indicate whether the participant was vaccinated (V) or not (N). The code to generate this figure can be found in https://doi.org/10.57745/KJGOYZ. CrI, credibility interval; HPV, human papillomavirus.

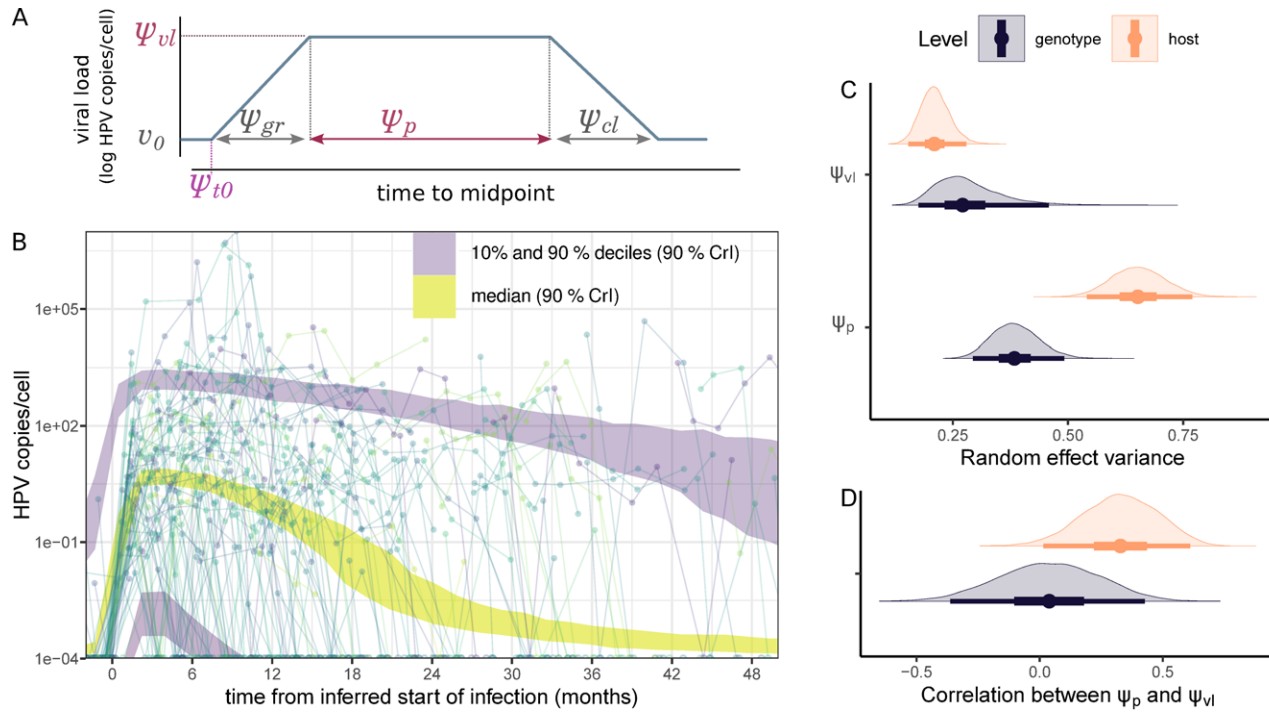

**Fig 4. Population predictions of viral load kinetics in HPV infections.** (A) Representation of the six parameters governing the descriptive models: plateau virus load $\psi_{vl}$, time of the infection $\psi_{t0}$, growth phase duration $\psi_{gr}$, plateau duration $\psi_p$, clearance phase duration $\psi_{cl}$, and initial viral load $v_0$. (B) The lines and points show the observed kinetics. The shaded areas show the population quantiles of the viral load through time simulated from the posterior distributions, with the median in yellow and the 10% and 90% deciles in grey (for all three, shaded areas are the 90% CrI). The time alignment was done using the posterior median time of infection. (C) Standard deviation of the random effects for the virus load and the infection duration related to the heterogeneity among virus genotypes (in black) or among hosts (in orange). (D) Correlation between these two random effects. The line represents the 95% quantiles, the thick box the interquartile range, and the dot the median of the posterior distribution. The code to generate this figure can be found in https://doi.org/10.57745/KJGOYZ. CrI, credibility interval; HPV, human papillomavirus.

many of our follow-ups are censored, meaning that we only observe a stable viral load and interpreting these with a two-phase model is difficult.

Taking these points into account, we developed a piecewise-linear mixed-effects model [36,37] in which we described virus dynamics with five population-level parameters capturing an increasing slope, a plateau, and a declining slope (Fig 4A and the methods). To model the variability between infections, both in terms of infection duration and plateau virus load, we introduced one random effect for the host and another for the HPV genotype. Intuitively, this means that instead of invoking 76 completely independent parameter values to capture the heterogeneity among our participants, we constrain this heterogeneity by modelling it as normally distributed (with a mean and variance estimated from the data), such that observations from one participant are partially informative of those from another participant. The same is true for the 13 HPV genotypes we observe. Our virus kinetics model assumed that infections within coinfected participants are independent. In order to compensate for the uncertainty arising from the censored follow-ups, we used a Bayesian inference method to fit our model. This allowed us to incorporate some prior knowledge from the literature, especially regarding infection duration [6]. As further detailed in the Methods, it also allowed us to maximise the exploitation of the information in the data, especially the type of follow-up censoring to estimate the time of infection ($\psi_{t0}$ in Fig 4A). For each of the parameters governing viral kinetics, the difference between the prior and the posterior distributions illustrates to what extent this prior information shapes the inferred value (Fig F in S1 Supplementary Materials).

The resulting model parameter estimates provided us with a description of the per-participant variability in normalised virus load values and exhibited wider uncertainty in censored follow-ups (Fig 3). According to the population average of the mixed effect model (Fig 4B), infections lasted for a median duration of 16 months [95% credibility interval (CrI) from 13 to 20], with a plateau lasting 14 months [95% CrI from 11 to 18] with a median of two virus genome copies per cell [95% CrI from 0.6 to 10] (respectively $\mu_p$ and $\mu_{vl}$ in Tables H and Fig F in S1 Supplementary Materials).

A striking feature of our resulting model is that most of the variance in infection duration originated from host differences (74% of the total explained variance for that parameter, with a 95% CrI from 61 to 84, $\Psi_p$ in Fig 4C). As a contrast, for the plateau virus load, heterogeneity between virus genotypes was comparable to that of the hosts ($\Psi_{vl}$ in Fig 4C), with more than two $\log_{10}$ differences between the most extreme genotypes (Fig G in S1 Supplementary Materials). At the host level, there was a positive correlation between the plateau duration and the plateau viral load (median correlation of 0.32 with a 95% CrI from 0.01 to 0.62, Fig 3D). This was not the case at the genotype level.

We explored the sensitivity of our results to variations in the most delicate parameter, namely that of the hyperprior on the random effects governing infection duration. When the random effect variance increased, the correlation between the plateau duration and plateau viral load vanished. Furthermore, this increase in variance also strongly affected the estimated duration of doubly censored follow-ups (Fig H in S1 Supplementary Materials). However, the effect was less pronounced for the other follow-ups. The advantage of constraining the random effect variance is that it forced the model to use knowledge from other follow-ups to inform doubly censored follow-ups, while avoiding imposing a strict upper bound to the infection duration. Additionally, when simulating data sets with a similar structure to ours using our default priors, i.e., performing a prior predictive check, we obtained infection durations consistent with earlier studies (Fig P in S1 Supplementary Materials).

Another sensitivity analysis we performed consisted in varying the number of negative visits required to consider the redetection of a genotype as a reinfection. When increasing the default value of two visits to three, we dropped from ten to four reinfections events in the data set, but this had a limited effect on the results (Fig I in S1 Supplementary Materials).

Finally, to validate the general shape of the virus kinetics, we compared our model to a more classical one assuming only an exponential increase followed by an exponential decrease of the viral load [38]. This simpler model did not perform well in modelling the data. In particular, contrarily to our model with a plateau, it was not able to correctly capture the longer stage with a high viral load of the longest infections, as can be seen in the highest decile of our posterior predictive check (Fig J in S1 Supplementary Materials).

## Immune kinetics

Characterising the virus load dynamics in HPV infections allowed us to investigate the temporal dynamics of the local immune response described earlier (Fig 1). For this, we used the same method to jointly analyse the time series of local concentrations of cytokines and counts of cervical leukocytes with a Bayesian piecewise-linear mixed-effects model aligned with the infection start and end dates inferred from the virus load data (Fig 5). This time, we used only two slopes and no plateau in the model for the immune effectors. We opted for a more parsimonious model because the two reasons that led us to hypothesise the presence of a plateau for the virus load were absent. Neither the raw data nor prior mathematical models supported the existence of a stage structure in the immune response.

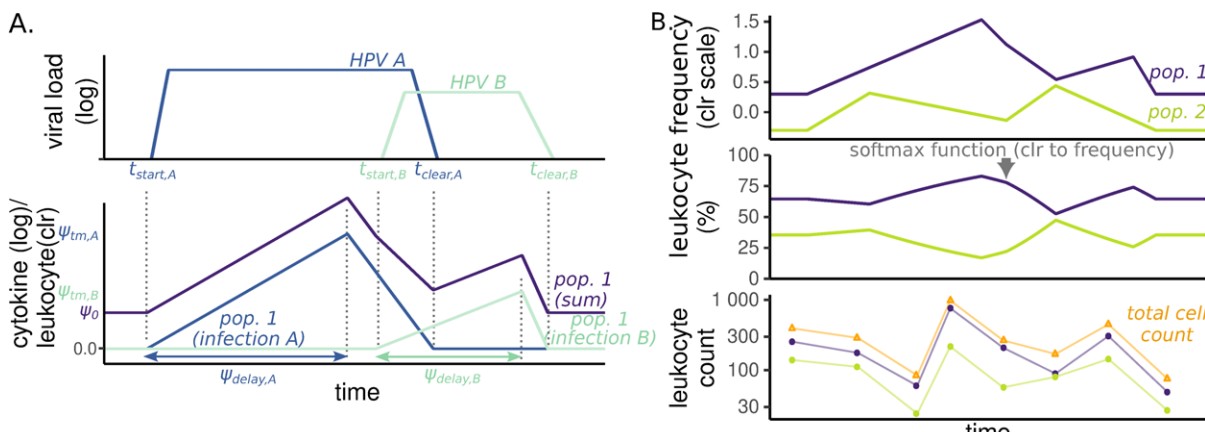

**Fig 5. Modelling local immune dynamics during HPV infections.** (A) Illustration of the parameters governing the descriptive models for a hypothetical case with two genotypes (*A* in dark blue and *B* in light blue) infecting an individual. For each infection, we have virus kinetics (top) and the immune dynamics model (bottom) is governed by four parameters: the peak of the response $\psi_{tm}$, the time to the peak $\psi_{delay}$, the beginning of the infection $t_{start}$, and its end $t_{clear}$. The first 2 are estimated, whereas the last 2 originate from the viral kinetics model. At the individual level, the effects of each infection are summed (in purple), assuming a basal level $\psi_0$. For the cytokines, this trajectory reflects the log of the normalised concentration. The trajectory can also decrease throughout the infection (i.e., $\psi_{tm} \in \mathbb{R}$). (B) Illustration of the additional steps for the FCM model. We consider the same two virus genotypes *A* and *B* and two hypothetical cell populations: 1, whose trajectory computation is illustrated in panel A, and 2, generated in the same way, with the same $t_{start}$, and $t_{clear}$, but different $\psi$s. For the FCM, the frequencies of each cell population are considered on the centred log-ratio (clr) scale. To compare our model to the count data, the clr trajectory is transformed back into frequencies using the *softmax* function and then multiplied by the total number of observed cells in the sample (see the Methods). The code to generate this figure can be found in https://doi.org/10.57745/KJGOYZ. FCM, flow cytometry; HPV, human papillomavirus.

Many study participants were coinfected. Therefore, the immune response we measured at a given time point resulted from the response to all the ongoing HPV infections, which also rendered a three-slope model less relevant than for the (genotype-specific) virus load. In the analysis, we chose to sum the effects of each infection at the individual level, which can already generate more variable patterns than in our virus dynamics model (Fig 5A). Fig 6 shows the individual trajectories of each of the immune variables for ten randomly selected participants.

For the cytokines, we fitted this model to the log of the normalised concentrations. At the population level, we did not observe a strong consistent increase or decrease in cytokines concentration in the cervical area, as illustrated by the peak sizes in Fig 7B or by the raw posterior and prior distributions in Fig K in S1 Supplementary Materials. The analysis of the patterns of between-individual variation indicated strong correlations of baseline values between two pairs of cytokines. CCL20 and CXCL10 had a median correlation of 0.47 [95% CrI from 0.22 to 0.63], and IL-17A and IFNγ had a median correlation of 0.53 [95% CrI from 0.30 to 0.64] (panel A of Fig M in S1 Supplementary Materials). Similar trends were observed for the peak value random effects but with weaker intensity (panels A and B of Fig M in S1 Supplementary Materials).

In the same model, we also fitted the count data of cervical leucocytes, under the assumption that the frequency of each cell cluster would increase at some point during the infection. As further detailed in the Methods, this required two additional steps to take into account that these data summed to 100%, i.e., are compositional (Fig 5B). At the population level, the frequency of some cell clusters peaked at similar times during an infection (Fig 7A and Fig L in S1 Supplementary Materials). The first group consisted of clusters I and II (granulocytes and NK cells) and appeared to be associated with the innate immune response. This group increased in proportion rapidly after the onset of infection, although with a weakly significant effect (median FC of cluster I compared to baseline frequency of 1.55 with a 95% CrI from

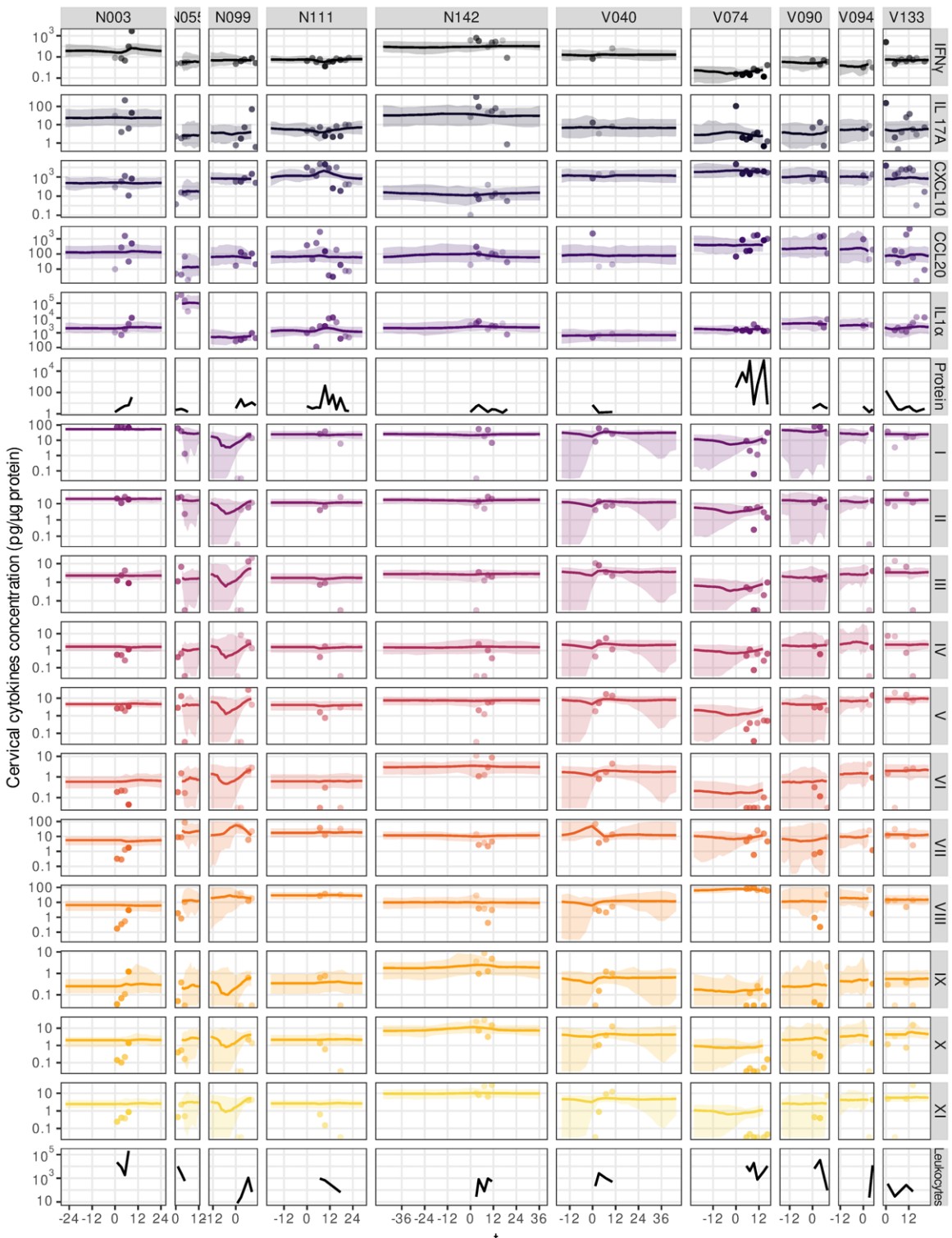

**Fig 6. Local immune dynamics during HPV infections for ten randomly selected participants.** Each row represents a local immune variable, and each column represents one participant. The dots indicate the observed data, the lines indicate the trajectory of the posterior median value, and the shaded area the 95% CrI from the bootstrapped posterior distribution. We also show the total protein concentration and total number of leukocytes as indicators of the sample quality for the cytokines assay and the FCM, respectively. The code to generate this figure can be found in https://doi.org/10.57745/KJGOYZ. CrI, credibility interval; FCM, flow cytometry; HPV, human papillomavirus.

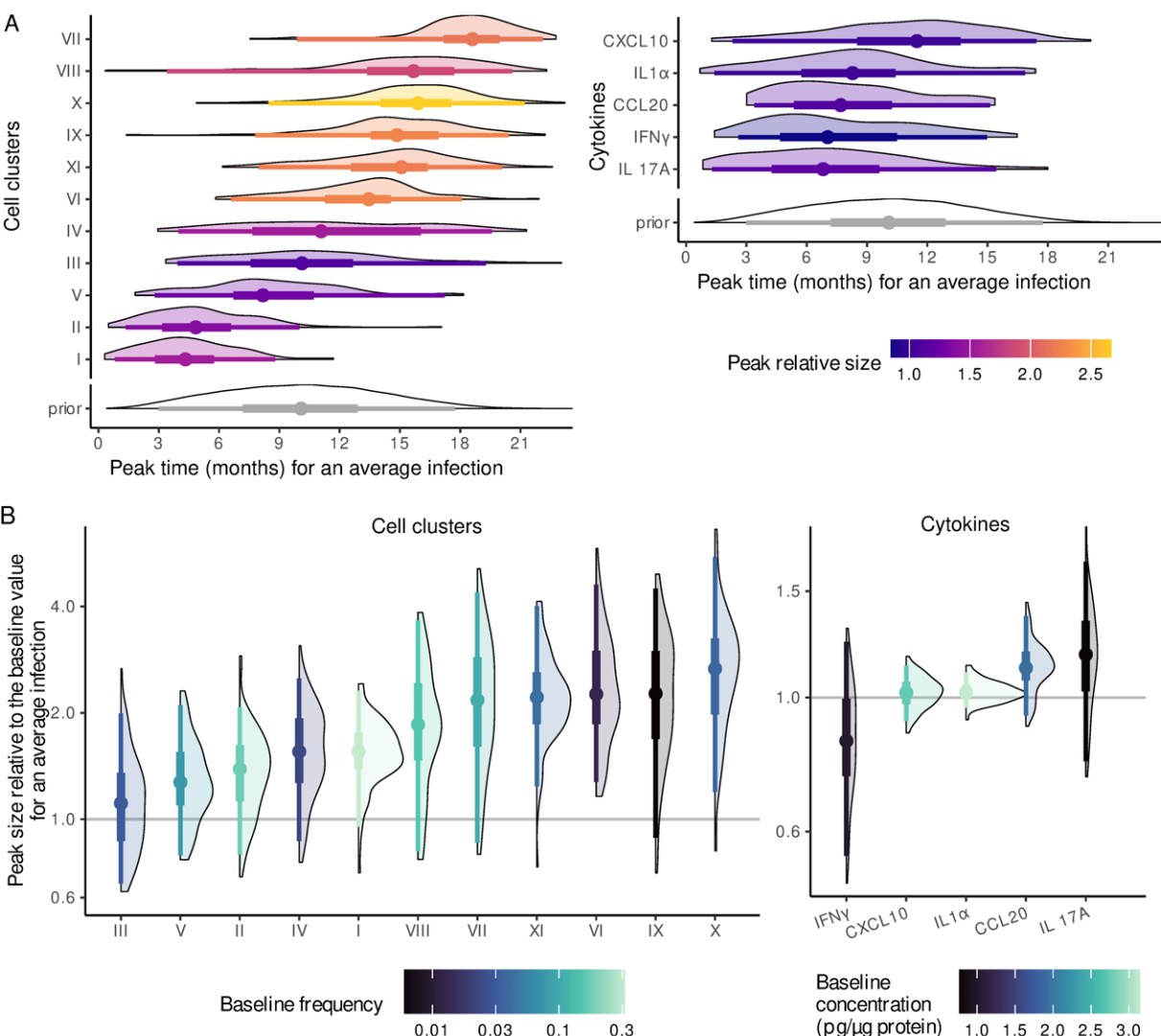

**Fig 7. Estimates of the key metric for local immune dynamics during the HPV infections.** (A) Aggregated posterior distribution of the median peak time of each immune variable, compared to the prior distribution. The x-axis was scaled to the estimated average infection duration, i.e., 16 months (with a 95% CrI of 13 to 20). The colour indicates the median peak fold change compared to the baseline value. (B) Distribution of the median leukocyte frequency or cytokine concentration fold change compared to the baseline, at the time of the peak. The colours indicate the value of the baseline frequency. The vertical line represents the 95% quantiles, the thick box the interquartile range, and the dot the median of the bootstrapped posterior distribution. The code to generate this figure can be found in https://doi.org/10.57745/KJGOYZ. CrI, credibility interval; HPV, human papillomavirus.

0.95 to 2.3, Fig 7B). Another group, which appears to be associated with the adaptive immune response (clusters VIII, VI, XI, X, and IX) exhibited a later peak. Within this group, the strongest relative increase was for CD4 T cells (cluster X, with a median FC of 2.7 with a 95% CrI from 1.2 to 5.5), while cluster XI (CD8 T cells) and cluster VI also significantly increased in frequency. There was weak evidence for an increase in the proportion of cluster VII (median FC of 2.2 with a 95% CrI from 0.9 to 4.3) with a median delay of 1.5 months [95% CrI from 0.2 to 8.1 months] before clearance, for a median infection, i.e., lasting around 16 months (Fig 7B). Furthermore, the host random effects controlling this cluster were strongly correlated with those of cluster VIII, both in the baseline frequency and in the peak amplitude (median

correlation coefficient of 0.54 [95% CrI from 0.21 to 0.77]; panel B of Fig M in S1 Supplementary Materials).

### Linking HPV within-host fitness and immunity

Finally, we analysed the virus within-host fitness in light of the dynamics of local immunity. To this end, we compared the cumulative viral burden, summarised as the area under the log of the virus load curve (area under the curve, AUC), with the main features of the immune response, namely serum antibodies, local cytokines, and local leukocytes.

First, we quantified circulating IgG and IgM antibodies that specifically target six of the genotypes that were also followed by qPCR (HPV16, HPV18, HPV31, HPV35, HPV52, and HPV58). This amounted to 56 infections from 38 participants in our data set, for which we could analyse the genotype-specific serological response to a focal infection. Although none of the vaccinated participants carried a focal infection by vaccine genotypes HPV16 or HPV18, they showed a higher average IgG titer (with a mean scaled titer difference of 1.3 [95% confidence interval [CI] from 0.7 to 2.0]), but no difference in IgM titer (Fig 8A). We did not detect significant correlations between the viral load AUC and the seroconversion status, the reference being the absence of seroconversion during the infection (Fig 8B). Similarly, we found no strong correlation between the mean antibody titer and the viral load AUC (Fig 8C).

Second, we calculated the correlation between the AUC of viral load and the mean value of each local immune variable, monitored during the 158 infections (Fig 8D). The only significant result was that the mean concentration of CXCL10 was negatively correlated with the viral load AUC (median correlation coefficient of −0.18 and 95% CrI from −0.31 to −0.03). Furthermore, we observed a trend for the cell populations that had a later peak in infection to have their mean frequency more positively correlated with the viral load AUC (Fig 8D).

## Discussion

While HPV infections in young adults typically do not persist for more than two years [4–7], the processes and mechanisms of HPV clearance remain elusive. Thanks to the combination of dense monitoring, high-quality data, and statistical models, we offer original insights into the dynamic interaction between HPV virus load and local immunity, from the onset of the infection to its clearance.

### HPV viral load kinetics in non-persisting infections

Building on an existing mathematical model [29] and on prior knowledge on infection duration [7], we show that HPV infections can be summarised in three phases: a short growth phase, a long plateau phase, and, for most of them, a short clearance phase. In particular, we show that this model outperforms a simpler one without the plateau phase. Our work is limited to studying the path to chronic infections because participants were not followed after 24 months of infection by the same HPV (only four of the 76 infected women in the cohort were followed while infected for 18 months or more, Fig E in S1 Supplementary Materials). However, having on-site visits every two months provided an unprecedented resolution to capture the natural history of these infections.

Given the substantial amount of heterogeneity among infections, powerful statistical techniques were required to isolate this general pattern. A Bayesian framework allowed us to use informative priors when the data contained little information. This was the case for the duration of infection, especially for the doubly censored follow-ups, and to a lesser extent, for the hyperpriors of the random effects governing the heterogeneity in infection duration and the plateau duration. The other parameters could be well estimated by the model, according to

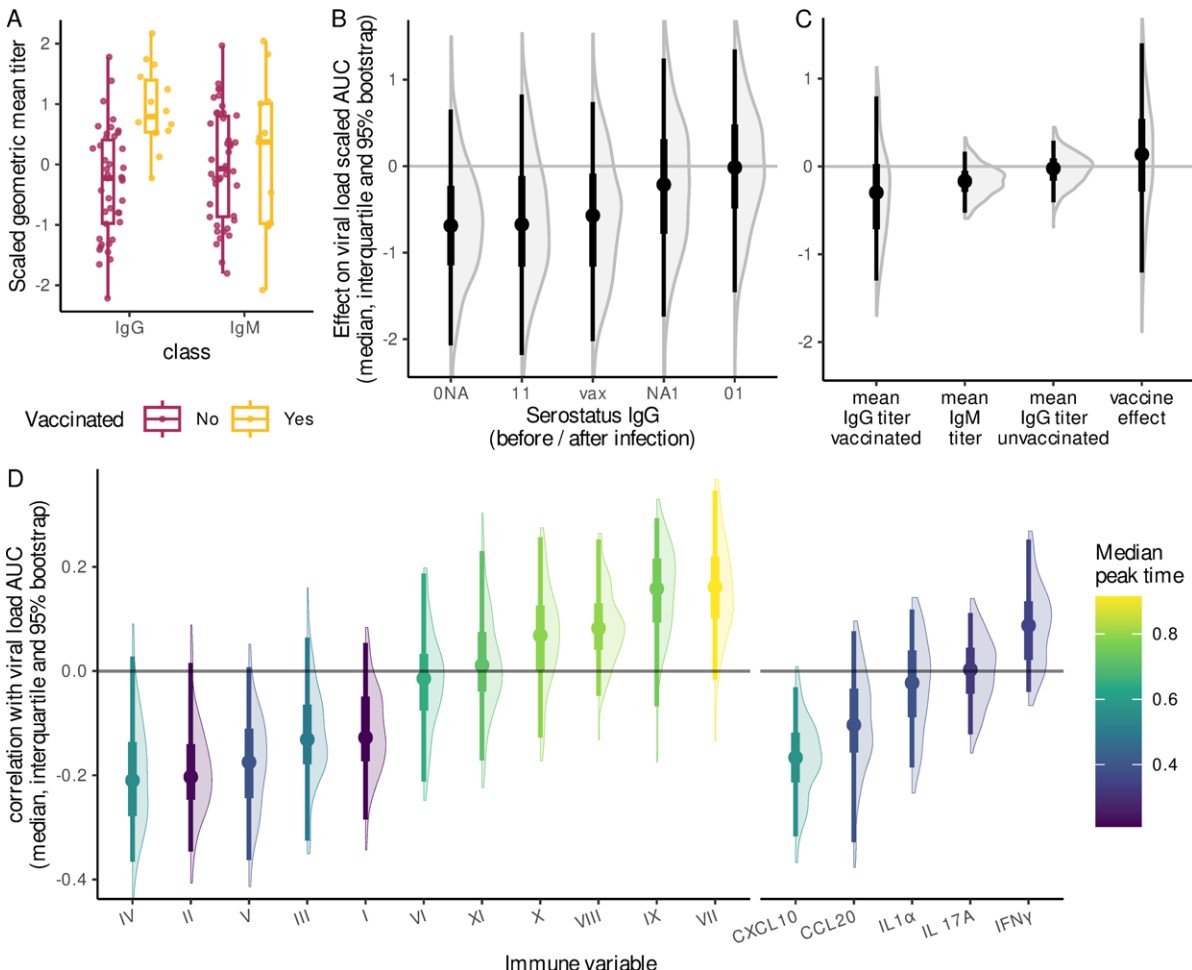

**Fig 8. Associations between HPV infection intensity and the immune response.** (A) Geometric mean IgG and IgM titers during the infection for vaccinated participants (yellow) or not (red). Antibodies were only measured in 56 infections (see the main text). (B) Regression coefficients of a linear model between IgG serostatus and viral load AUC. For each infection, "0" indicates seronegativity, "1" seropositivity in at least one sample, and "NA" missing information (because of censored follow-ups). The first value indicates the status before the infection, and the second, the status after the infection, whence the notations "0 0," "0 1," or "NA 1." All vaccinated individuals were seropositive and treated as a separate class ("vax"). The reference is "0 0," i.e., individuals who never seroconverted during the infection. (C) Regression coefficients of a linear model between the geometric mean antibody titer measured during infection and viral load AUC. (D) Distribution of the bootstrapped correlation between the individual effect on viral load AUC and the individual effect on local immune variable mean value during the infection. The panel colours indicate the median peak time (also shown in Fig 7A) for an average infection. In A, the three components of the box and whisker represent the median, interquartile range, and 95% range. The other panels represent the median aggregated posterior distribution from 500 bootstrapped regressions from the viral kinetics fit (the dot), the interquartile range (the thick line), and the 95% CrI (the thin line). Immune response variables and viral load AUC were centred and reduced to compare the effects. The code to generate this figure can be found in https://doi.org/10.57745/KJGOYZ. CrI, credibility interval; HPV, human papillomavirus.

which both HPV genotype and host contribute to viral load variance at the plateau phase, whereas host differences play a dominant role in explaining infection duration variance. Interestingly, there was a positive correlation between the host-specific effect on infection duration and the host-specific effect on plateau viral load, which is consistent with previous observations made specifically on HPV16 [39–41].

## Associations between HPV infections and local immunity patterns

Given the little prior knowledge, we studied both innate and adaptive immune responses. Focusing on lymphoid immunity and using state-of-the-art unsupervised clustering

techniques allowed us to highlight a dominance of granulocytes, which is consistent with earlier studies [42]. More originally, our approach revealed a strong association between TCRγδ cells and HPV infections, with the identification of at least three distinct subpopulations. We also found a lower proportion of CD4 T cells (cluster X) in individuals with a focal HPV infection than in uninfected individuals, suggesting that this population may not play a major role in controlling the infection locally. These results complement previous studies on cervical smears that did not include antibodies specific to TCRγδ in their panels [26,42].

Another strength of unsupervised clustering is that it allows us to detect unexpected immune cell populations of interest. This is the case of cluster VIII, which we had difficulties to identify. Based on its expression pattern, we hypothesise that it could contain ILCs or progenitors, but not ILC3 given the lack of correlation between this cluster and IL-17A (Fig 2) [43]. Even if we are not able to label it accurately, we know that this population is more frequent in focal HPV infections and deserves further experiments to be characterised in detail.

Most of our knowledge of the local immune response in the cervical area comes from biopsies. For ethical reasons, these cannot be performed in a cohort such as this one given the number of visits and the asymptomatic nature of the infection. However, future studies could further improve the characterisation of the local immunity in cervical smears by taking advantage of recent technologies. For example, cytometry by time of flight (CyTOF), i.e., mass cytometry, could be particularly appropriate given that it relies on isotopes and would not be affected by the autofluorescent nature of the keratinocytes. This technique also allows the inclusion of many more markers in the analysis than the flow cytometry we used. Other novel single-cell omics approaches that can accommodate delicate samples with few immune cells could also yield a more accurate picture of the cervical immune response in the context of an HPV infection.

## Insights from the temporal dynamics of local immunity

Our hierarchical model enabled us to describe the kinetics of local immune cells during a typical HPV infection. Clusters associated with the innate immune response, especially granulocytes (cluster I), increased in frequency earlier in the infection. This supports the hypothesis that the immune system can detect an HPV infection early on and react to it. Conversely, clusters associated with the adaptive immune response increased in frequency later.

This timing data also helped us to better identify some clusters. For example, the three that we associated with TCRγδ cells (clusters III, VII, and IX) had different kinetics, which points to different mechanisms controlling their presence in the cervix. Cluster III did not show any increase during infections, and its baseline frequency was positively correlated with that of cells associated with the innate immune response, in particular granulocytes (cluster I).

Clusters VII and IX peaked at different times (1.5 and 4.7 months before infection clearance respectively to a median infection). One hypothesis could be that populations from clusters VII and IX are part of the same TCRγδ cellular population, VII being a potentially mature, or differentiated, form of population IX. Independently of the exact interpretation, the relative timing of the peaks suggests that the former may be more associated with the resolution of the local inflammation.

Regarding the cytokines, the strongest signal was that infections with higher culmulative viral burden were significantly associated with decreased CXCL10 concentrations. This cytokine is stimulated by IFNγ [44], which was not significantly correlated with infection duration. Furthermore, we observed a trend for cell clusters associated with an innate immune response to be more frequent in infections with smaller viral load AUC. We also found weak evidence for a slight increase in CCL20 during infection (median fold change of 1.12 [95% CrI from

0.93 to 1.37]) and a slight decrease in IFNγ (median fold change of 0.85 [95% CrI from 0.55 to 1.24]).

## Immunity and HPV infection early or late clearance

By putting together our analyses of the local and systemic immune response, and even though we do not have the statistical power to demonstrate detailed mechanistic aspects, we can formulate an interpretation of how HPV infection clearance could unfold.

A striking pattern in our results is that some hallmarks of the innate immune response, such as CXCL10, are associated with milder infections. Furthermore, more expectedly, we find that immune cell populations associated with the adaptive immune response have their peak later in the infection. One possibility to reconcile these observations is to hypothesise that a fraction of HPV infections manage to evade the innate immune response, in particular, the interferon response pathway, thereby establishing a more persistent infection. As a consequence, we find an association between innate immune effectors such as CXCL10 and milder infections. Conversely, infections with higher cumulative burden trigger the adaptive immune response and can be associated with leucocyte populations such as cluster VII. We identify this cluster as a more mature or differentiated population of TCRγδ cells, potentially infiltrated from the blood to the mucosa.

Challenging this interpretation with existing data is difficult because most studies are cross-sectional, which makes the HPV status assessment more uncertain (as illustrated by the occurrence of "singletons" in our data). Furthermore, previous studies often do not analyse the data at the genotype level but rather compare samples positive for any HPV genotype in their assay to samples negative for all the HPVs in their assay.

Bearing in mind these limitations, our results are in line with several elements from the literature. First, natural immunity to HPV infections is known to be weakly protective [45], which is consistent with the idea that innate immunity may sometimes clear the infection. Furthermore, a recent study found HPV clearance to be associated with lower IL1-β/CXCL10 ratios [22]. This is consistent with our observation that HPV–negative samples are associated with higher concentrations of CXCL10. This could further be interpreted as a more general role for the local immune environment that would modulate, before and after infection, the level of cytokine expression. This is also consistent with our finding that the negative correlation between the viral load AUC and the CXCL10 concentration was mostly driven by the baseline value of the latter.

One limitation of our interpretation is that our model may have overlooked rapid immune response dynamics, potentially misattributing temporal variation to a heightened baseline. Therefore, focusing specifically on the early or later stages of the infection, i.e., the growth and decay phases in our three-slope model, could yield valuable insights regarding the mechanistic aspects of HPV clearance.

## Limited vaccine-induced cross-immunity

An earlier cross-sectional analysis of this cohort found that vaccination status was associated with differences in HPV detection risk, HPV coinfections, and genotype prevalence [46]. In our longitudinal analysis, we did not detect any "breakthrough" infection, which we define as the focal infection of a vaccinated participant by one of the vaccine genotypes. Therefore, any potential effect of vaccination on infection duration or plateau viral load would necessarily come from cross-immunity.

At the immunological level, we detected slight associations between vaccination status and immune cell populations in the cervical area. In particular, a population that may correspond

to monocytes (cluster V) was more frequent in samples from vaccinated women (Table D in S1 Supplementary Materials).

Our analysis of the AUC of the virus kinetics did not detect any significant association with HPV vaccination status. However, we observed a trend for the participants being already seropositive for a genotype before the infection by this genotype (whether by vaccination or prior infection) to have a smaller viral load AUC than those who never seroconverted (Fig 8B). The median effect on the centred reduced viral for unvaccinated was −0.7 with a 95% CrI from −2.2 to 0.8.

Vaccinated individuals were all already seropositive before infection, even for genotypes not targeted by the vaccine, and there was weak evidence that they exhibited higher IgG titers during the infection. Besides a potential lack of specificity in our assay, this could be interpreted as a mild cross-protective mechanism [47,48]. In particular, there was weak evidence for the mean IgG from vaccinated individuals and IgM titers during the infection to be negatively correlated with the viral load AUC, which is consistent with the hypothesis of a biological mechanism. Therefore, natural immunity may protect against HPV infection, but not enough to prevent it entirely. Note that the serological status was missing for some HPV genotypes, which means that systemic immunity could have a larger effect on the viral kinetics.

Performing similar analyses on "breakthrough" infections could be particularly interesting to identify potential immune escape strategies from the virus.

## Perspectives

Our study provides original insights on HPV non-persisting infections. It focused on a specific population, namely young women, most of whom were students, living in a country with a largely socialised healthcare system. This is unlikely to have much effect on the generalisability of our within-host dynamics results, especially given our modelling with random effects. Furthermore, our results are consistent with the limited data from the literature, for example, in terms of the type of leucocytes we detect in the cervical smears (especially the dominance of granulocytes [42]), or in terms of HPV genotype prevalence in this age class. However, extending this approach to other populations could help clarify questions at the population level, both on the host and the virus side. Furthermore, being able to generate less censored follow-up would improve the accuracy of the inference by decreasing the importance of the prior distributions.

Our analyses were limited by many unknowns regarding HPV life history of the associated immune response. Based on our results, investigating markers more related to myeloid immune cells appear to be a promising strategy to improve our understanding of HPV clearance. Another line of investigation could consist in building on existing within-host dynamics models to estimate more biologically relevant parameters, such as the "burst size" of infected cells or the killing rate by the immune response [29]. As mentioned above, more recent technologies such as CyTOF or single-cell multiomics could also help refine our understanding of the local immune response.

Both the vaginal microbiota [49] and the immune response vary through a menstrual cycle [50] with, for instance, fewer immunoglobulins during the luteal phase and, conversely, more IL-1. Furthermore, different vaginal community state types have been differentially associated with the risk of HPV detection [34] but investigating how menstrual cycles interact with the course of HPV infection dynamics would require denser follow-ups. This also applies to HPV dynamics because, although dense, our follow-up has a 4-month uncertainty window around "singletons," i.e., HPV detection at a unique visit. Studies with denser follow-ups are required to better understand whether these events correspond to short productive infections.

Cohort participants were followed for a mean duration of 290 days. Long-term follow-up of these women could yield valuable insights, especially thanks to new developments in HPV full genome sequencing [51]. For example, it could help estimate the prevalence of latent infections [52] but also further monitor the dynamics of chronic infections [53]. Finally, it would provide critical data on HPV within-host evolution, which could be associated with cancer development since HPV16 found in cervical cancers has a particular genomic signature with less variation in the E7 gene [54].

More generally, HPVs represent an ideal system to better understand the sometimes tenuous frontier between non-persisting and chronic viral infections [55].

## Materials and methods

### The PAPCLEAR clinical study

The PAPCLEAR study is a monocentric cohort study that was designed to follow $N = 150$ women in the context of HPV infections (see ref. [28] for details about its protocole). Its inclusion criteria were to be between 18 and 25 years old, to live in the area of Montpellier (France), to be in good health (no chronic disease), not to have a history of HPV infection (e.g., genital warts or high-grade cervical lesion), and to report at least one new sexual partner over the last 12 months. Enrolment was independent of HPV infection and vaccination status. Given the lack of prior knowledge of HPV infections, the cohort size was chosen to maximise the number of inclusions and follow at least 75 infected women until infection clearance.

Participants were enrolled by putting up posters and handing out leaflets at the main STI detection centre (CeGIDD) within the University Hospital (CHU) of Montpellier (France) and in the Universities of the city. To increase enrolment, posters were also hung at bus stops near the CHU.

The inclusion visit (V1) was performed by a gynaecologist or a midwife at the CeGIDD outside operating hours. After an interview, biological samples were collected by the clinician during a gynaecological consult, after which a nurse collected blood samples and participants filled in a detailed questionnaire.

The second visit (V2) was scheduled four weeks later to inform the participants of the result of a screening test for cervical lesions (see also ref. [46]). Depending on their HPV status at the inclusion visit (see below), participants were oriented to the HPV–negative or to the HPV–positive arm of the study.

Participants in the HPV–negative arm, for whom no alphapapillomavirus was detected at V1, came back for on-site visits every four months. If an HPV was detected at one of the following visits, they switched to the HPV–positive arm. HPV–negative participants not infected by month 32 of the study were not followed anymore.

Participants in the HPV–positive arm had on-site visits every two months until clearance or chronification of the infection. Chronicity was defined as 24 months of infection by the same HPV genotype. Clearance of a specific infection was defined as two successive visits without any detection of the focal genotype.

The longitudinal cohort was complemented by a cross-sectional cohort, which was originally designed to enrol the same number of women, i.e., 150, who would only perform the inclusion visit (V1) and the results visit (V2). The inclusions began in November 2016 and, due to the COVID-19 pandemic and the associated strain on hospital services, the study was prematurely ended in September 2020. This led to the loss of follow-up of two participants and 110 non-inclusion in the cross-sectional cohort. Overall, we followed $N = 189$ women, including 149 (78.8%) in the longitudinal cohort, with a total of 974 on-site visits with gynaecological consults (Fig O in S1 Supplementary Materials).

## Clinical samples description and processing

**Cervical smears.**  A cervical smear was collected at each of the on-site visits. The gynaecologist or midwife performed 2.5 turns before putting the brush into 20 mL Thinprep PreserCyt medium from Hologic (at V1) or 5 mL of fresh phosphate-buffered saline (PBS) medium (at the other visits).

Smears in PreserCyt medium were analysed by a trained pathologist of the CHU of Montpellier. Before that, 2 mL were sampled for HPV detection. Leftovers from the cytological analysis were also stored.

Smears in PBS were processed following a general protocol described in ref. [25]. The whole protocol was performed at 4°C. Within 2 h of collection, the smears were vortexed during 45 s with the cytobrush, which was then removed carefully to save as many cells as possible before adding 5 mL of Roswell Park Memorial Institute (RPMI) medium. The solution was then filtered in a new tube with a cell strainer (Fisherbrand Sterile Cell Strainers, 100 μm); 250 μL of the resulting solution were aliquoted for backup. The remaining solution was centrifuged for 10 min at 514 g. The resulting supernatant was stored at −80°C and the cell pellet was washed with 5 mL PBS at 514 g for 10 min. The supernatant was discarded, and cells were resuspended in 200 μL, and 20 μL of the resulting solution was aliquoted for HPV detection, while the rest was processed for flow cytometry staining.

**Ophtalmic sponges.**  As described in ref. [21], during the on-site visits, cervicovaginal secretions were collected by the gynaecologist or midwife using WeckCel sponges (Beaver-Visitec International) placed directly into the cervical os for 1 min. Sponges were then transferred into a Salivette (Sarstedt) device and centrifugated at 1,500 rpm for 5 min at 4°C after the addition of 175 μL of PBS. Supernatants were separated into 50 μL aliquots and stored at −80°C.

**Blood samples.**  During on-site visits, 20 mL of circulating blood was collected by a trained nurse for antibody titration and STI detection.

## HPV detection and genotyping

HPV detection and typing was performed on the sample originating from the cervical smear and resuspended in 200 μL of PBS as described above. From this, we extracted DNA using the QIAamp DNA mini kit (QIAGEN Inc.) following standard protocol for body fluids (spin control).

We first tested for the presence of alphapapillomaviruses using the generic DEIA test [56] from DDL Diagnostic Laboratory (Rijswijk, the Netherlands). We then used the LiPA25 typing [57], also from DDL Diagnostic Laboratory, on DEIA-positive samples. Both assays target the same amplicon of the L1 HPV gene.

Samples that were DEIA-positive and LiPA25 negative were amplified using the PGMY PCR [58] and sequenced using Sanger sequencing. Samples for which the sequencing did not yield a clear sequence, most likely because of coinfections, were labelled as "non-typable."

## Genotype-specific HPV qPCR

We quantified the number of genomic copies by qPCR for 12 HPV genotypes (HPV16, 31, 35, 39, 51, 52, 53, 56, 58, 59, 66, and 68) using the protocols and primers from ref. [35]. We also quantified the number of copies of HPV18 and of a human reference gene (albumin). The HPV18 forward primer was 5′-ACACCACAATACCATGGCG-3′, the HPV18 reverse primer was 5′-TTC AGTTCCGTGCACAGATC-3′, and the HPV18 probe was 5′-FAM-CAACACG\GCGACCCTA CAAGCTAC-HBQ1-3′. For the albumin primers, the details can be found in ref. [59].

All qPCRs were performed using SensiFAST Probe No-ROX Kit (MERIDIAN Bioscience) on LightCycler 96 and LightCycler 480 (Roche Diagnostics) for 384 wells plates. Cycles used for

all genotypes and albumin were 95˚C for 5 min and 40 two-step cycles with 95˚C for 10 s and 60˚C for 30 s. Each sample was run in triplicate with 2 μL of samples in a final volume of 10 μl. To create HPV standards, we used plasmids graciously provided by the international HPV reference centre from the Karolinska Instituet (https://www.hpvcenter.se). Each plasmid contained a full HPV genome for one of the 13 target genotypes. For the albumin standard, we used genomic female Human DNA (Promega). For each participant, we ran on the sample plate the qPCR for all samples, all genotypes previously identified by LiPa$_{25}$ and the albumin gene.

Virus load measures were taken in triplicates and we analysed the mean cycle threshold (Ct) value. HPV genotype-specific Ct values were normalised by twice that of the cellular gene, the albumin (known to be present at two copies per diploid genome).

## Cytokines quantification

We used the MSD U-plex Biomarker Group 1 (human) from Meso Scale Discovery (MSD, Rockville, Maryland, USA) to measure the concentration of five analytes identified in a previous study as being associated with HPV infection [21]: IFN-γ, IL-10, CCL20, IL-17A, and CXCL10. According to the manufacturer's instructions, we used 25 μL of specimen, which was obtained from the ophtalmic sponges as detailed above. Analyses were performed on a MESO QuickPlex SQ 120 reader.

To normalise cytokine quantitation, we measured total protein concentration in the same samples using the Invitrogen Qubit protein assay (Thermo Fisher Scientific) following the manufacturer's instructions.

To compute cytokine sample concentrations from the standard curves, we used a four-parameter logistic regression model implemented in the minpack.lm R package.

## Antibody dosage

IgG and IgM antibodies targeting L1 proteins of high-risk HPV types 16, 18, 31, 33, 35, 45, 52, and 58, as well as low-risk HPV-types 6 and 11 were analysed with a multiplex serology assay using beads coated with recombinant glutathione s-transferase (GST) fusions proteins. The assay procedure has been previously described in detail [60]. The serum samples were tested at a final dilution of 1:100 using an IgG and an IgM goat anti-Human secondary antibody. Seropositivity was defined based on standard definitions [61,62]. We chose to focus on circulating antibodies in the serum rather than on mucosal antibodies in vaginal secretions because the latter are less abundant, they originate from samples that are difficult to standardise, and their variability is still poorly understood [63].

## Flow cytometry

For the FCM analyses, within 3 h upon sampling, cellular suspensions were labelled with Live Dead red diluted 1/2,000 (Thermo Fisher) for 20 min in the dark, on ice. Cells were then washed with 2 mL of PBS (1,500 RPM, 5 min) and resuspended in 100 μL of PBS, before being transferred to a new tube containing dried antibodies (DURAClone, Beckman Coulter). No blocking step was undertaken before the labelling step. This custom antibody panel included anti-CD45 KRO (clone J33, IM2473U), anti-CD16 FITC (clone 3G8, B49215), anti-CD3 APC A750 (clone UCHT1, A94680), anti-CD4 APC A700 (clone 13B8.2, B10824), anti-CD8 PB (clone B9.11, A82791), anti-TCRγδPC5.5 (clone IMMU510, A99021), anti-CD69 PE (clone TP1.55.3, IM1943U), and anti-CD161 PC7 (clone 191B8, B30631).

Cells were incubated in the presence of antibodies for 15 min in the dark at room temperature, then washed with 2 mL of PBS (1,500 RPM, 5 min). Finally, cells were re-suspended in

250 µl of PBS 1% PFA and stored at 4°C until acquisition using a flow cytometer (Navios, Beckman Coulter).

From 631 acquired samples, 161 (25.5%) displayed visible pellets of red blood cells and were not analysed further.

## Unsupervised FCM analysis

For unsupervised analyses, raw LMD files were converted to FCS 3.0 using a custom R script. Samples with visible blood pellets were excluded from the analysis.

Raw files were cleaned using the PeacoQC R package, and then filtered for viable cells (Lived Dead Red⁻). Among the 470 samples without visible red blood cells (RBC) pellets, 87 (17.7%) did not pass this quality control step and we eventually analysed 362 samples.

Using the CATALYST pipeline and FlowSOM, which builds on a Self-Organizing Map (SOM), an unsupervised technique for clustering and dimensionality reduction, we first restricted the data set to $CD45^+$ cells [64,65].

We then clustered the cell populations using FlowSOM with 100 clusters and 20 metaclusters. The clustering was defined on all available samples, including both HPV–negative and HPV–positive samples. The following parameters were used to perform the clustering: SSC-A, FSC-A, CD45, CD3, CD4, CD8, TCRγδ and CD16. CD69 and CD161 were only used for differential expression analysis. The 20 metaclusters thus generated were manually merged into 11 cell populations (see Figs A and B and Table B in S1 Supplementary Materials).

A UMAP restricted to 150 cells per sample and on the same parameters used with FlowSoM clustering was performed to obtain a visual representation of the 11 cell population clusters.

We tested for differential abundances of the 11 cell populations between HPV–negative and focal HPV–positive samples using the diffcyt R package with the diffcyt-DA-edgeR method and including a random effect at the participant level. Differential expressions were calculated in a similar way using the diffcyt-DS-limma method. In both settings, adjusted $p$-values were obtained by applying a Benjamini–Hochberg procedure to correct for false discovery rate (FDR).

## Basic statistical analyses

All statistical analyses were run with R version 4.4.

In Table 1 and Table A in S1 Supplementary Materials, we used respectively Kruskal–Wallis rank sum test kruskal.test function for numeric variables and the chisq.test function for the binary and categorical variables from the TableOne R package.

In Fig 2A, we used the brms R package to build a multivariate model. The five cytokines or chemokines were considered as separate response variables. Each participant was assumed to have a varying intercept. We assumed correlations between varying effects. The only fixed effect was the HPV focal status. The multivariate model was implemented with the mvbind function brms. To estimate the bayesian $R^2$ in Table G in S1 Supplementary Materials, we used the loo_R2 function from the brms package.

In Fig 1B, we calculated correlation matrices with the lmer function (using the lme4 R package). Each cell population was considered as a response variable, with all five cytokines and chemokines as fixed effects, yielding 11 independent linear models. A random effect for the participant was also included in each model. The regression coefficient of each model is named "$\beta$" in the text.

## Bayesian hierarchical modelling

All kinetics were modelled using rstan v. 2.32.3 (Stan version 2.26.1). The script used and raw data are available on https://doi.org/10.57745/KJGOYZ. We used the settings described in

S1 Supplementary Materials in Table K for our models. The credible intervals were computed using the 2.5th and the 97.5th percentiles of the posterior distributions, i.e., with equally tailed intervals. In the figures, we systematically summarised the results by showing the median, interquartiles, and the 95% posterior distribution.

In the following, we describe the statistical modelling approach performed to infer the viral dynamics and the associated immune dynamics. To maximise the simplicity of the notations, $\mathcal{N}(\mu, \sigma)$ represents the normal distribution with mean $\mu$ and standard deviation $\sigma$, $\mathcal{N}(\mu, \Sigma)$, the multivariate normal distribution with variance-covariance matrix $\Sigma$, and $\mathcal{E}(\lambda)$ represents the exponential distribution with rate $\lambda$. Finally, we use the notation $v^{[n]}$ to represent a vector $v$ of length $n$.

**Virus load dynamics.** *qPCR data.* This analysis included all the participants who had at least three on-site visits and at least one positive virus load with a genotype-specific qPCR (see Fig O in S1 Supplementary Materials).

For some participants, for a given genotype, there were HPV–negative visits between HPV–positive visits. In the analysis, we assumed that HPV–positive visits separated by two or more consecutive negative visits (i.e., at least four months) corresponded to two different infection events. Overall, this generated ten additional infections.

The qPCR data was pre-processed using the following steps:

1. Samples with fewer than 100 copies of albumin and with HPV copies below the limit of detection (l.o.d.), i.e., three copies, were excluded;

2. Samples with HPV copies below the l.o.d. and at least 100 copies of albumin were treated as censored at the limit of quantification (l.o.q.), i.e., ten copies;

3. Samples with at least 100 albumin copies and HPV copies between the l.o.d. and the l.o.q. were considered interval-censored between one and ten copies;

4. Samples with fewer than 100 albumin copies and HPV copies above the l.o.d. were deemed unreliable, with the number of HPV copies per cell considered above $10^{-4}$ (following the standard limit of detection [40]); and

5. Samples with at least 100 copies of albumin and HPV copies above the l.o.q. were retained as is. The number of HPV copies per cell was then computed as the number of HPV copies divided by half the number of albumin copies.

**The model.** Because of the shape of the raw data and of previous mechanistic mathematical modelling of HPV genital infections [29], we assumed that virus load dynamics followed a pattern with an exponential growth phase, a plateau, and a rapid clearance phase. These dynamics were captured using five parameters (Fig 4A) describing the $\log_{10}$ virus load, i.e., the number of HPV copies per cell, over time $v(t)$, with $t$ the time since the beginning of the follow-up. Formally, our virus load dynamics can be written as follows:

$$v(t) = \begin{cases} v_0 & \text{if } t < \psi_{t0} \\ v_0 + \dfrac{\psi_{vl}}{\psi_{gr}}(t - \psi_{t0}) & \text{else if } t < \psi_{t0} + \psi_{gr} \\ \psi_{vl} & \text{else if } t < \psi_{t0} + \psi_{gr} + \psi_p \\ \psi_{vl} - \dfrac{\psi_{vl}}{\psi_{cl}}\left(t - \left(\psi_{t0} + \psi_{gr} + \psi_p + \psi_{cl}\right)\right) & \text{else if } t < \psi_{t0} + \psi_{gr} + \psi_p + \psi_{cl} \\ v_0 & \text{else} \end{cases} \quad (1)$$

where, as illustrated in Fig 4A, the parameters indicate the plateau virus load ($\psi_{vl}$), the date of infection ($\psi_{t0}$), the growth phase duration ($\psi_{gr}$), the plateau duration ($\psi_{p}$), and the clearance phase duration ($\psi_{cl}$). $v_0$ is the minimal viral load. All duration parameters are expressed per month, and $\psi_{vl}$ and $v_0$ are expressed as $\log_{10}$ (HPV copies/cell).

**Estimation of the infection date.** To facilitate model convergence, we computed the starting date of each infection ($\psi_{t0}$) differently depending on its follow-up censoring status. For each participant $i$, each genotype $g$, and each infection $j$:

1. When the follow-up was right-censored (i.e., the participant was not infected at inclusion but still infected at the last visit), we estimated $\psi_{t0}$ relatively to the observed infection time $\tilde{t}_{0,j}$ (middle of last negative and first positive sample), assuming the informative prior $\psi_{t0,j} \sim \mathcal{N}(\tilde{t}_{0,j}, 1)$.

2. When the follow-up was left-censored (i.e., the participant was infected at inclusion but at least one negative observation in the end), we estimated $\psi_{t0}$ relatively to the observed clearance time $\tilde{t}_{clear,i}$ (middle of the last positive and the first negative sample), assuming the informative prior $\psi_{t0,j} \sim \mathcal{N}(\tilde{t}_{clear,j} - \psi_{gr} - \psi_{p} - \psi_{cl}, 1)$.

3. When the follow-up was complete (i.e., at least one negative observation at inclusion and one negative observation in the end), we estimated $\psi_{t0}$ relatively to the observed midpoint of infection $\tilde{t}_{mid,j}$, i.e., the mean date of all the genotype-specific positive samples from an individual. We the assumed the informative prior $\psi_{t0,j} \sim \mathcal{N}(\tilde{t}_{mid,j} - \psi_{gr} - \psi_{p}/2, 1)$.

4. When the follow-up was doubly censored, we also estimated $\psi_{t0}$ relatively to the observed midpoint of infection $\tilde{t}_{mid,j}$. We estimated its prior distribution, using simulations where we assumed a uniform probability of inclusion and dropout throughout the infection, and based on known infection duration distributions [6]. This resulted in a prior distribution of $\psi_{t0,j} \sim \mathcal{N}(\tilde{t}_{mid,j} - \psi_{gr} - \psi_{p}/2, 7)$.

These modelling choices resulted in assuming an absence of interaction between viral load time series within coinfected hosts.

Finally, for follow-ups where only one negative sample was observed at the end or at the beginning of the infection instead of two, we accounted for the risk of false-negative measurement in 10% of the cases. More specifically, we assumed $\psi_{t0}$ to be a mixture of the case with a true negative in 90% of the cases and a false negative in 10% of the cases, the latter leading to a different follow-up censoring status and an adapted computation of $\psi_{t0}$.

**Parameters decomposition.** To estimate the other model parameters, we made several assumptions regarding random effects and transformation of variables. In particular, we performed a log transform for the duration parameters, and an exponential transform for the viral load to constrain it to be above $v_0$.

Mathematically, for a given participant $i$, genotype $g$, and infection $j$, we estimated the following parameters:

$$\log(\psi_{p_{i,j,g}}) = \log(\mu_p) + \eta_{p_i} + \rho_{p_g} + \delta_{reinf_j}\xi_p \tag{2A}$$

$$\log(\psi_{vl_{i,j,g}}) = \log(\mu_{vl}) + \eta_{vl_i} + \rho_{vl_g} + \delta_{reinf_j}\xi_{vl} \tag{2B}$$

where $\mu$ indicates a fixed effect, and $\begin{bmatrix} \eta_p \\ \eta_{vl} \end{bmatrix} \sim \mathcal{N}(0, \Omega_\eta)$ are the host random effects,

$\begin{bmatrix} \rho_p \\ \rho_{vl} \end{bmatrix} \sim \mathcal{N}(0, \Omega_\rho)$ are the genotype random effects. $\Omega_\eta$ and $\Omega_\rho$ denote the variance-covariance matrices. Using the Cholesky decomposition, these were decomposed as the product of a lower triangular matrix, respectively $L_\eta$ and $L_\rho$ with its conjugate transpose, and scaled elementwise by the outer product of a scale vector, with respectively $\omega_\eta^{[2]}$, $\omega_\rho^{[2]}$ representing the variance.

In equation system 2, $\delta_{\mathrm{reinf}_j}$ represents a dummy variable with value 1 when the infection is the second one with this same genotype in a given participant. Note that $j$, which can take values 0 or 1, is nested in $g$ because it describes whether the second genotype is identical to the first one or not. The reverse is not true since genotype $g$ is assumed to have the same effect across hosts. $\xi_p$ and $\xi_{vl}$ denote the reinfection (fixed) effects on the plateau duration and plateau viral load, respectively.

We did not include random effect parameters for the growth and the clearance phases of the dynamics because they were often unobserved, either due to the censoring of the follow-up or due to the short duration of those phases compared to the sampling frequency.

All the priors used and their units are listed in Table L in S1 Supplementary Materials.

**Immune dynamics: Cytokines and FCM.** Similarly, but separately, for the cytokines concentrations and leukocyte counts, we assumed an additive response due to each HPV infection, and denoted the amount of the immune marker (whether a concentration or a count) by $v(t)$, with $t$ the time since the beginning of the participant's follow-up. In what follows, we will first detail the computation of $v(t)$, and then explain its fit to the observations.

**Computation of $v(t)$.** We first decompose $v(t)$ as a baseline value $\psi_0$, and a specific response to each infection, $z(t)$. Mathematically, for one infection, the value of the response against genotype $g$, at time $t$ of an immune marker $k$ can be written as follows:

$$z_{k,g}(t) = \begin{cases} 0 & \text{if } t < t_{\mathrm{start}} \\ \dfrac{(t - t_{\mathrm{start}})\psi_{\mathrm{tm}}}{(t_{\mathrm{clear}} - t_{\mathrm{start}})\psi_{\mathrm{delay}}} & \text{else if } t < t_{\mathrm{start}} + (t_{\mathrm{clear}} - t_{\mathrm{start}})\psi_{\mathrm{delay}} \\ \psi_{\mathrm{tm}} - \dfrac{(t - t_{\mathrm{start}} - (t_{\mathrm{clear}} - t_{\mathrm{start}})\psi_{\mathrm{delay}})\psi_{\mathrm{tm}}}{(t_{\mathrm{clear}} - t_{\mathrm{start}})(1 - \psi_{\mathrm{delay}})} & \text{else if } t < t_{\mathrm{clear}} \\ 0 & \text{else if } t > t_{\mathrm{clear}} \end{cases} \tag{3}$$

where $\psi_{\mathrm{tm}}$ is the relative increase or decrease due to this infection, and $\psi_{\mathrm{delay}}$ is the relative timing of the peak (as a proportion of the total infection duration). We used the posterior distribution of the virus dynamics to define the beginning and end of each infection, noted respectively $t_{\mathrm{start}}$ and $t_{\mathrm{clear}}$, which also correspond to the beginning and the end of the slopes (Fig 5A).

For infections with more than one genotype, we assumed an additive effect of each genotype on the immune response, assuming that the effect of each infection is independent of the others. We also assumed a baseline level for each immune marker $k$ in each participant $i$, $\psi_{0,i,k}$. Therefore, by summing over all the genotypes present $G$, we obtain the following equation for the temporal dynamics of the amount of a marker in an individual:

$$v_{i,k}(t) = \psi_{0,i,k} + \sum_{g=1}^{G} z_{i,k,g}(t) / n_{\mathrm{coinf}}^{\gamma_k}$$

$$= \psi_{0,i,k} + n_{\mathrm{coinf}}^{\gamma_k - 1} \sum_{g=1}^{G} z_{i,k,g}(t) \tag{4}$$

We corrected for this addition of several kinetics by a factor $(n_{\mathrm{coinf}})^{\gamma_k}$, where $n_{\mathrm{coinf}}$ is the total number of infections in a participant during the follow-up. $\gamma$ was allowed to vary for each cytokine, but for identifiability reasons, we used one $\gamma_{\mathrm{FCM}}$ for all cell clusters. Intuitively, if $\gamma = 0$ each infection effect adds up, if $\gamma = 1$ immune response is the average of each infection, and if $\gamma > 1$ coinfections are expected to result in a milder response than a single infection. In the case of a single infection ($n_{\mathrm{coinf}} = 1$), the amount of a marker $v_{i,k}(t)$ is simply the sum of its baseline value $\psi_{0,i,k}$ and the deviation triggered by the infection $z_{i,k,g}(t)$.

**From $v(t)$ to the likelihood.**  We then took into account the specificities of our samples to convert $v(t)$ into the expected dynamics of our observations.

For the cytokines, $v(t)$ represents the $\log_{10}$ of their normalised concentrations.

Due to the specificity of the tissue analysed, the flow cytometry data were compositional, meaning that their space is a simplex $\mathcal{S}^K$, with $K$ the number of leukocytes populations. This imposed two constraints on our model.

First, we used a multinomial-logistic link between the leukocyte frequencies $v'(t)$ and the two-slope dynamic $v(t)$. This means that $v(t)$ can be seen as a centered log-ratio (clr) scale of the frequencies. The clr function converts the frequencies $v' \in \mathcal{S}^K$, with $K$ the number of leukocytes clusters, into an unbounded vector $v \in \mathbb{R}^K$ with the equation: $v = \left[ \log \frac{v'_1}{g(v')}, \ldots, \log \frac{v'_K}{g(v')} \right]$, where $g(v')$ is the geometric mean of $v'$.

Second, since we used a clr transformation, we had to remove one degree of freedom in our equation to keep parameters identifiable. This was done by restricting the sum of the baseline level:

$$\sum_{k=1}^{K} \psi_{0,k} = 0 \tag{5}$$

Similarly, to keep peak sizes identifiable, we assumed that an observed decrease in frequencies for one cell type can only be due to the increase in frequencies of other cell types (Fig 6B). Therefore, we assumed that $\psi_{tm,k} > 0$ for all $k \in \mathrm{FCM}$.

**Parameters decomposition.**  As for the virus kinetics, we introduced random effects and performed transformations of some variables to perform the parameter inference. In particular, we used a logit transform to constrain the peak/drop time to a proportion of the duration of the corresponding HPV infection, and a log transform for the FCM peak values to constrain them to be positive.

Mathematically, for each individual $i$, each HPV infecting genotype $g$, and each cytokine (or cell cluster) $k$, we estimated the following parameters:

$$\psi_{0_{i,k}} = \mu_{0_k} + \eta_{0_{i,k}} \tag{6A}$$

$$\psi_{\mathrm{tm}_{i,k,g}} = \mu_{\mathrm{tm}_k} + \eta_{\mathrm{tm}_{i,k}} + \rho_{\mathrm{tm}_{k,g}} \quad \text{for the cytokines,} \tag{6B}$$

$$\log(\psi_{\mathrm{tm}_{i,k,g}}) = \mu_{\mathrm{tm}_k} + \eta_{\mathrm{tm}_{i,k}} + \rho_{\mathrm{tm}_{k,g}} \quad \text{for the FCM,} \tag{6C}$$

$$\mathrm{logit}(\psi_{\mathrm{delay}_{i,k,g}}) = \mathrm{logit}(\mu_{\mathrm{delay}_k})(1 + \tau t_{\mathrm{infscaled}_{i,g}}) + \eta_{\mathrm{delay}_i} \tag{6D}$$

where the $\mu$s indicate fixed effects. $\tau$ represents the correction factor for the infection duration $t_{\mathrm{inf\ scaled},i,g}$ so we expect the peaks early in the infection to appear proportionally later in shorter infections, and vice versa.

Furthermore, following the sum-constraint of compositional data (Eq 5), for each sub-parameter, we also have: $\sum_{k=1}^{K \in \text{FCM}} \mu_{0,k} = 0$ and $\sum_{k=1}^{K \in \text{FCM}} \eta_{0_{i,k}} = 0$.

In equation system 6, we assumed three random effects associated with each individual $i$. The first one, $\eta_{0_i}^{[15]}$, represents the impact on the baseline value, with $\eta_{0,i} \sim \mathcal{N}(0, \Omega_{\eta_0})$. This vector is the concatenation of the effect of each cytokine and each leukocyte population: $\eta_{0_i} = (\eta_{0_{\text{IFN}\gamma,i}}, \ldots, \eta_{0_{\text{CCL20},i}}, \eta_{0_{1,i}}, \ldots, \eta_{0_{\text{X},i}})$. Joining them allows us to include a correlation between the leukocytes frequencies and the cytokine concentrations, which is biologically expected. Indeed, immune variations between women can be expected to be driven by physiological differences in body mass index (BMI), stress, or nutrition; all of which are known to affect immunity. Therefore, these could be jointly affecting the baseline level of the local immune markers that we measured. The vector is of length 15 because we have five cytokines, and ten degrees of freedom for the 11 leukocyte populations due to the sum constraint.

The other two are denoted $\eta_{\text{tm}_i}^{[16]}$ (the random effect associated with the peak/drop value) and $\eta_{\text{delay}_i}^{[1]}$ (the random effect associated with the peak times shift). They are jointly correlated through a vector formed by the concatenation of $(\eta_{\text{tm}_i}, \eta_{\text{delay}_i})^{[17]} \sim \mathcal{N}(0, \Omega_{\eta_{\text{tm}}})$. To avoid inflating the total number of parameters to estimate, we use the same value $\eta_{\text{delay}_i}$ for all cytokines and cell clusters of participant $i$, which is therefore of length 1. The underlying assumption is that the peak order is the same for all participants, following the fixed effect value, and can only be shifted earlier or later in the infection for each participant $i$.

There is one random effect associated to the HPV genotype $g$, $\rho_{\text{tm}_g}^{[16]}$, which is associated with the peak/drop value, $\rho_{\text{tm}_g} \sim \mathcal{N}(0, \Omega_{\rho_{\text{tm}}})$.

As for the viral kinetics model, $\Omega_{\eta_0}$, $\Omega_{\eta_{tm}}$ and $\Omega_{\rho_{\text{tm}}}$ denote the variance-covariance matrices and were decomposed using the Cholesky decomposition as the product of a lower triangular matrix, respectively $L_{\eta_0}$, $L_{\eta_{\text{tm}}}$ and $L_{\rho_{\text{tm}}}$, with their conjugate transpose, and scaled elementwise by the outer product of a scale vector, respectively $\omega_{\eta_0}$, $\omega_{\eta_{\text{tm}}}$ and $\omega_{\rho_{\text{tm}}}$ representing variance.

**Likelihood computation.**   Since the five cytokine measurements originated from the same sample, we assumed that the $\log_{10}$ of each of their normalised concentrations, denoted $c_i(t)$, follow a multivariate normal distribution with a variance-covariance matrix $\Sigma_{\text{cyt}}$:

$$c_i(t) \sim \mathcal{N}(v_{i,k}(t), \Sigma_{\text{cyt}}) \quad \text{for } k \text{ in cytokines} \tag{7}$$

This matrix was also decomposed using the Cholesky decomposition, as the product of a lower triangular matrix, $L_{\text{cyt}}$ with its conjugate transpose, and scaled elementwise by the outer product of a scale vector $\sigma_{\text{cyt}}^{[2]}$.

For the FCM, we first converted $v(t)$ into frequencies $v'(t)$ using the inverse of the multinomial logistic transformation, also called softmax function, i.e., $v' = \left[ \frac{e^{v_1}}{\sum_{k=1}^{D} e^{v_k}}, \ldots, \frac{e^{v_K}}{\sum_{k=1}^{K} e^{v_k}} \right]$.

We then used the initial cell count data, denoted $f_k(t)$ to compute the likelihood of our model. Following the results of ref. [66], we assumed a Beta-Binomial distribution to capture the counts from our compositional analyses, with a log-error parameter $\pi_k$. In summary, the idea is that although the Beta-Binomial, by construction, is not constraining frequencies to sum to 1, it offers more flexibility for the error parameter which is a vector specific to each cell population $k$. The sum-constraint is imposed from the prediction model itself with the softmax function, which constrains the prediction vector $v'(t) \in \mathcal{S}^K$ i.e., in a simplex summing to 1.

Mathematically, this can be written as follows:

$$f_{i,k}(t) \sim \text{BetaBin}(F, e^{\pi_k}\nu'_{i,k}(t), e^{\pi_k}(1 - \nu'_{i,k}(t)))\text{for } k \text{ in FCM} \tag{8}$$

Where $F$ is the total number of cells measured in a given sample.

The prior distributions for each parameter are described in S1 Supplementary Materials in Table M for FCM, Table N for cytokines, and Table O for the joint variance-covariance matrix.

### Regressions between immune variables and viral kinetics

**Correlations involving local immunity.** In a third step, we computed the average level of each immune variable $k$ throughout the infection, which we denoted $\alpha_k$, as well as the individual average area under the viral load curve (AUC) which we denoted $\beta_{\text{auc}}$, $\beta_{\text{vl}}$, and $\beta_p$. Taking advantage of the double random effect structure, for each individual $i$, we computed these metrics extracting the fixed effects and host random effects:

$$\alpha_{i,k} = \eta_{0_{i,k}} + \frac{\eta_{\text{tm}_{i,k}}}{2n_{\text{coinf}}^{(\gamma_k-1)}} \quad \text{for the cytokines} \tag{9A}$$

$$\beta_{p_i} = \exp(\eta_{\rho_i} + \mu_p) \tag{9B}$$

$$\beta_{\text{vl}_i} = \exp(\mu_{\text{vl}} + \eta_{\text{vl}_i}) \tag{9C}$$

$$\beta_{\text{auc}_i} = \beta_{\text{vl}_i} \times \beta_{p_i} \tag{9D}$$

To compute the FCM mean values, we first simulated the individual trajectory by discretising the time from infection to clearance in 100 time points and then estimated the average of those time points. We used the individual average $\psi_{0,i,k} = \mu_{0_k} + \eta_{0_{i,k}}$, $\psi_{\text{tm}_{i,k}} = (\mu_{\text{tm}_k} + \eta_{\text{tm}_{i,k}})n_{\text{coinf}}^{(\gamma_{\text{FCM}}-1)}$ and $\text{logit}(\psi_{\text{delay}_{i,k}}) = \text{logit}(\mu_{\text{delay}_k}) + \eta_{\text{delay}_{i,k}}$.

From these estimates, for each $k$, we then computed the Pearson correlation coefficient between $\alpha_k$ and $\beta_{\text{AUC}}$.

**Correlations involving serological data.** The HPV genotype-specific serostatus for each sample was based on standard cutoff values [61,62]. We used the following criteria to stratify the individuals as a function of their IgG and IgM serostatus. First, participants were considered as seropositive initially if we could measure at least one seropositive sample before 15 days after the onset of the infection; they were considered as seronegative initially if the last sample observed before or the first sample observed after the onset of the infection was seronegative; we considered the data as "NA" (not available) otherwise. Second, participants were considered as "seropositive after the infection" if at least one sample after the onset of the infection was seropositive and as "seronegative after the infection" if there was at least one sample after the viral clearance and all these samples were seronegative. Otherwise, the data was labelled as not available ("NA"). All vaccinated individuals were IgG seropositive before the infection and, therefore, treated as a separate class. In what follows, we considered the "NA" cases before and after infection as a specific third category.

We fitted a linear model with random effects on the HPV genotype and host, using the viral load AUC as the response variable and the IgG serostatus before and after the infection, as an explanatory variable. We also fitted a similar mixed effects model, with viral load AUC as a response variable, and with the log of the maximal observed IgG and IgM titer after the onset of the infection as explanatory variables.

These models were fitted with the R package brms with the default priors, and sampling scheme detailed in Table K in S1 Supplementary Materials.

## Parameters bootstrapping

We repeated the immune dynamics fits and the correlations' computations 500 times: starting from one sample in the HMC chain of the viral kinetics model, we estimated the immune response dynamics. We then used one sample in the HMC chain of this model to compute the correlation analysis.

The distributions of the parameters and of the correlations were obtained from these aggregated posterior bootstraps. In the results, they are referred to as bootstrap intervals.

## Supporting information

**S1 Supplementary Materials. Supplementary Information.**
(PDF)

## Acknowledgments

The authors thank all the study, the CeGIDD and hospital staff from the CHU of Montpellier for their commitment and help.

The authors acknowledge the ISO 9001 certified IRD i-Trop HPC (member of the South Green Platform) at IRD Montpellier for providing HPC resources that have contributed to the research results reported within this article (bioinfo.ird.fr and www.southgreen.fr). The authors acknowledge further support from the Centre National de la Recherche Scientifique, the Institut de Recherche pour le Développement, the Fédération Hospitalière Universitaire InCH of Montpellier.

## Author Contributions

**Conceptualization:** Michel Segondy, Ignacio G. Bravo, Nathalie Boulle, Carmen Lía Murall, Samuel Alizon.

**Data curation:** Nicolas Tessandier, Baptiste Elie, Christian Selinger, Thomas Beneteau, Tsukushi Kamiya, Bastien Reyné, Marie-Christine Picot, Carmen Lía Murall, Samuel Alizon.

**Formal analysis:** Nicolas Tessandier, Baptiste Elie.

**Funding acquisition:** Carmen Lía Murall, Samuel Alizon.

**Investigation:** Nicolas Tessandier, Vanina Boué, Massilva Rahmoun, Claire Bernat, Sophie Grasset, Soraya Groc, Marine Bonneau, Christelle Graf, Christophe Hirtz, Jacques Reynes, Vincent Tribout, Tim Waterboer, Michel Segondy, Nathalie Boulle.

**Methodology:** Nicolas Tessandier, Baptiste Elie, Vanina Boué, Christian Selinger, Massilva Rahmoun, Claire Bernat, Soraya Groc, Anne-Sophie Bedin, Thomas Beneteau, Marine Bonneau, Christelle Graf, Nathalie Jacobs, Tsukushi Kamiya, Marion Kerioui, Julie Lajoie, Imène Melki, Jean-Luc Prétet, Bastien Reyné, Géraldine Schlecht-Louf, Mircea T. Sofonea, Olivier Supplisson, Chris Wymant, Vincent Foulongne, Jérémie Guedj, Christophe Hirtz, Marie-Christine Picot, Jacques Reynes, Vincent Tribout, Édouard Tuaillon, Tim Waterboer, Michel Segondy, Ignacio G. Bravo, Nathalie Boulle, Carmen Lía Murall, Samuel Alizon.

**Project administration:** Samuel Alizon.

**Resources:** Jean-Luc Prétet, Vincent Foulongne, Christophe Hirtz, Marie-Christine Picot, Vincent Tribout, Édouard Tuaillon, Tim Waterboer, Michel Segondy, Nathalie Boulle.

**Software:** Nicolas Tessandier.

**Supervision:** Samuel Alizon.

**Validation:** Nathalie Jacobs, Imène Melki, Géraldine Schlecht-Louf, Jérémie Guedj, Samuel Alizon.

**Visualization:** Nicolas Tessandier, Baptiste Elie.

**Writing – original draft:** Nicolas Tessandier, Baptiste Elie, Samuel Alizon.

**Writing – review & editing:** Nicolas Tessandier, Baptiste Elie, Bastien Reyné, Géraldine Schlecht-Louf, Mircea T. Sofonea, Olivier Supplisson, Chris Wymant, Ignacio G. Bravo, Carmen Lía Murall, Samuel Alizon.

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
