## [Editor Report · Decision Letter 0]

10 Jun 2024

Dear Dr Alizon, 

Thank you for submitting your manuscript entitled "Viral and immune dynamics of human papillomavirus genital infections in young women" for consideration as a Research Article by PLOS Biology.

Your manuscript has now been evaluated by the PLOS Biology editorial staff, as well as by an academic editor with relevant expertise, and I am writing to let you know that we would like to send your submission out for external peer review.

Once your full submission is complete, your paper will undergo a series of checks in preparation for peer review. After your manuscript has passed the checks it will be sent out for review. To provide the metadata for your submission, please Login to Editorial Manager (https://www.editorialmanager.com/pbiology) within two working days, i.e. by Jun 12 2024 11:59PM.

Kind regards,

Melissa

Melissa Vazquez Hernandez, Ph.D.

Associate Editor

PLOS Biology

---

## [Decision Letter · Decision Letter 1]

31 Jul 2024

Dear Samuel,

Thank you for your patience while your manuscript "Viral and immune dynamics of human papillomavirus genital infections in young women" went through peer-review at PLOS Biology. Your manuscript has now been evaluated by the PLOS Biology editors, an Academic Editor with relevant expertise, and by three independent reviewers, being reviewer #2 Rob J. de Boer.

As you will see in the reports, all reviewers are positive about the relevance of the work, and they offered several suggestions for improvements. Reviewer #1 recommended that you conduct flow cytometry or single-cell RNA sequencing, as well as repeating the analysis of associations between cytokines and cell clusters. Reviewer #2 suggested that you improve the readability of the paper. Reviewer #3 questioned to which extent the immune markers can explain the viral clearance. After discussion with the Academic Editor, we strongly suggest that you conduct the flow cytometry experiments recommended by Reviewer #1 if samples are available. However, we think that the scRNAseq is not necessary for the study. 

**IMPORTANT - SUBMITTING YOUR REVISION**

*Resubmission Checklist*

*Published Peer Review*

*PLOS Data Policy*

*Blot and Gel Data Policy*

Sincerely,

Melissa

Melissa Vazquez Hernandez, Ph.D.

Associate Editor

PLOS Biology

REVIEWERS' COMMENTS

Reviewer #1: Overview

In the submitted article, Viral and immune dynamics of human papillomavirus genital infections in young women, Tessandier and Elie et al. detail the establishment of the PAPCLEAR observational human cohort study enrolling HPV- and HPV+ women and utilized samples collected between 2016-2020 to study non-persistent HPV infections. From 189 participants, authors collected temporal viral titer (RT-PCR) and immune measure (clustered flow cytometry; cytokine; antibody) data, which were incorporated into a series of Bayesian models of viral dynamics and associated immune responses over time. The major findings of the study include i) identification of immune clusters from cervical smears that differ between HPV- and HPV+ participants; ii) generation of a three-slope HPV non-persistent infection model lasting approximately 14 months with a 12-month plateau bookended by rapid infection/clearance dynamics; iii) identified associations between CXCL10 and γδ T cells with viral clearance. This is a well-written manuscript, which does a thorough job explaining the modeling techniques used and establishes a novel cohort to answer critical questions related to common HPV infections. The clarity and manuscript would be improved with a few updates described below:

Main Comments

Some of the model priors, such as infection duration and midpoint, are informed by the empirical data, but the rationale for hyperparameters of the decomposed terms is not fully explained. Since posterior parameter estimates are sensitive to the selection of priors, how would the authors quantify the robustness of the model (e.g. Sensitivity analysis)?

For the viral load kinetic model presented in Figure 4, is there a simpler baseline model (from either the literature or from the authors' own experience) with fewer parameters, over which the proposed model shows improvement? Similarly, is there a quantifiable improvement of the three-step over the two-step model or is this only displayed visually in Figure S6?

In Figure 2B, the authors look for associations between cytokines and cell clusters, though these data appear to combine both infection and uninfected samples. It would be informative to repeat this analysis after splitting the data into focal and non-focal subsets as this may improve resolution of cell-cytokine associations specific to focal infections. 

In a subset of participants in which sampling prior to, during, and following HPV infection are available, it would be informative to perform a functional flow cytometry assay by stimulating collected cells with HPV peptides or inactivated whole virus then determining cytokine production by ICS - i.e. determine the cellular source of the cytokines. However, I appreciate that this may not be possible with the limited cells obtained from cervical sampling. To get around this issue (and if additional funds are available), the authors should consider single-cell RNA sequencing as another option to determine the cellular source of cytokines, which would also detect some of the additional cell types (e.g. ILCs) hypothesized to be present in the Discussion.

Minor Comments

1. 'Longitudinal cohort' is redundant as 'cohort' implies longitudinal sampling is performed

2. Please include a complete flow cytometry gating scheme with representative images of each cell cluster in supplementary materials

3. Table S1 is missing (labeling begins at Table S2)

4. In Fig 1B and C, please align Groups I-XI

5. Data from the multivariate model is not present in Fig. 1C (line 70)

6. Please order and align cytokines in Fig 2A and 2B to match

7. Fix typo in line 85 from "0.96" to "0.096"

8. Figure 8 legend references "Fig 7C", which does not exist

9. Was there a blocking step included in the flow cytometry staining methods?

10. Please define all model terms (e.g. ψPmax and ξPj are undefined)

11. Fig S10 color scale difficult to visualize; could include a light grid

12. Fig S12 appears to have missing values for 'x'

13. It is unclear if cytokines and antibodies were measure from peripheral blood or from cervical samples. Please indicate

14. After 'samples with visible pellets' are removed, how many remain for analysis?

15. Table S5: it would be informative to include the 'Explained Variance' metric from the brms R package

16. Line 212, there is a space needed after TCRγδ 

Reviewer #2: 

This is an impressive paper analyzing unique and precious longitudinal data on HPV infectious in a large group of healthy volunteers. The data provides a first look into the typical duration of such infections, the frequency with which they occur and the type of immune responses they invoke. The data is skillfully analyzed with appropriate piecewise models and fitted using a mixed effect population approach using Bayesian approaches in Stan. Because HPV infections occur so frequently, because they have very important health consequences, because they are only cleared after a very long period, and because the analysis is cutting-edge, this manuscript was an exciting read. 

That being said my major recommendation would be to improve the readability of the paper. Keep the general audience in mind. The methods are complicated and novel, and the writing style is quite technical, e.g., consistently speaking of clusters rather than cell types, technical abbreviations like "clr", and sometimes complicated wording like "which is consistent with the hypothesis of a biological mechanism" (line 281). I admire the non-supervised definition of cellular phenotypes, but the reader would be helped when the biological interpretations of the 11 clusters are repeated more frequently.

It would also be good to expand the biological discussion: 

- What is the effect on vaccination on infections with other HPV genotypes?

- Why do you think it takes so long to clear an infection while the innate responses start quite early?

Minor recommendations

I am confused that the initial viral load, v_0, is identifiable parameter because the time of infection is not known. Wouldn't the Viral load growth duration define both the time of infection and the initial viral load? 

I would provide a reference to FlowSOM and provide a short sentence explaining its algorithm 

line 50 We could also notably identify => We identified 

line 145 Why would baseline values of cytokines be correlated?

In Fig 7 The shading of the bullets indicates total protein or total cell numbers. Although it is excellent that this information is provided, it quite difficult to see

line 543 Still notationwise, in equation system => In equation system

Reviewer #3: 

This is a very interesting article on the natural history and immune dynamics of HPV infections, based on a unique cohort of healthy women sampled every two months over a relatively long time period. The paper is well written (perhaps at times a bit difficult to read; but this is to some extent inevitable given the complexity of the topic) and the analysis seems sound. I particularly liked the idea of exploring "the frontier between acute and chronic infections " which I expect to be of broader interest. 

I have the following suggestions:

-It would be interesting if the authors could discuss the generalizability of their findings beyond the specific population included. 

- The authors fit the virus load dynamics with a phenomenological/statistical model consisting of an increasing slope, a plateau, followed by a decreasing slope. It would be good if they could provide some mechanistic justification for this assumption (as it is interesting but rather unusual). In the discussion they refer to a previous paper ("Building on an existing mathematical model [32] and prior knowledge on infection duration [7], we show that HPV infections can be summarised in three phases… ") as a potential explanation but it would be good to have this reasoning earlier and more explicitly in the manuscript. Similarly, it was not fully clear to me how ""This assumption regarding the shape was motivated by the high frequency of left- and right-censored follow-ups …". Please clarify. 

- The same applies to some extent also to the two-slope model for the immune dynamics. Here, it would be additionally good if the authors could argue to what extent this qualitatively matches the observed dynamics. 

-It would be good if the authors could provide a bit more interpretations/speculations/context for the different immune markers they find to be associated with HPV infection (or infection characteristics); e.g. what does the association of TCR-gamma-delta cells and CXCL10 concentration with HPV infection imply for our understanding of HPV immune dynamics. 

- Could these immune markers (TCR-gamma-delta cells and CXCL10 ) be corroborated in women with incident infections or in women clearing HPV infections (i.e. do they increase/decrease upon infection or clearance in the way predicted by their overall associations with HPV status across individuals).

-Also it was not fully clear to what extent the different immune markers can explain the clearance after the end of the plateau. 

-I would rather omit the sentence "Our linear models identified positive correlations between the 83 concentration of IFNγ and the frequency of CD8 T cells (cluster XI, with a regression coefficient, see the Methods, β = 0.11), although with a p-value of 0.96." given the p-value. 

-Given the large number of different often non-standard analyses performed, it would be good if the code could be provided in a repository (e.g. githbub or similar). 

- The sentence "Our virus kinetics model assumes that the infections are independent. Therefore, we aligned each time series to the time to the observed midpoint of the corresponding infection " was not fully clear to me. Please clarify.

---

## [Decision Letter · Decision Letter 2]

23 Oct 2024

Dear Dr Alizon,

Thank you for your patience while we considered your revised manuscript "Viral and immune dynamics of human papillomavirus genital infections in young women" for publication as a Research Article at PLOS Biology. This revised version of your manuscript has been evaluated by the PLOS Biology editors, the Academic Editor, and two of the original reviewers.

Based on the reviews and our Academic Editor's assessment of your revision, we are likely to accept this manuscript for publication, provided you satisfactorily address the following data and other policy-related requests.

IMPORTANT - please attend to the following:

a) While your current Title is OK, it's more idiomatic in English to say "genital human papillomavirus infections." Also we wonder if you could make it more informative? Two options that we suggest are "Longitudinal viral and immune data on human papillomavirus genital infections in young women reveal the dynamics of persistence and clearance" and "Viral and immune dynamics of genital papillomavirus infections in young women with high temporal resolution"

b) We note that you currently have a structured Abstract. Please could you change it to an unstructured format (in both the manuscript and the Editorial Manager metadata) in accordance with our style.

c) Please address my Data Policy requests below; specifically, we need you to supply the numerical values underlying Figs 1AB, 2, 3, 4BCD, 5B, 6, 7AB, 8ABCD, S1, S2, S3, S4, S5, S6, S7, S8, S9, S10, S11, S12, S13, S14, S16, either as a supplementary data file or as a permanent DOI’d deposition. I note that you already have plans for a CNRS Dataverse deposition. Please complete this deposition with the data and code needed to recreate the Figures, as we will need to check compliance before accepting for publication.

d) Please cite the location of the data clearly in all relevant main and supplementary Figure legends, e.g. “The data underlying this Figure can be found in S1 Data” or “The data underlying this Figure can be found in https://zenodo.org/records/XXXXXXXX

e) Please make any custom code available, either as a supplementary file or as part of your data deposition. I see that you mention plans to deposit these in Zenodo; again, we will need to check compliance.

We expect to receive your revised manuscript within two weeks. 

*Published Peer Review History*

*Press*

Sincerely,

Roli Roberts

Roland G Roberts PhD

Senior Editor

PLOS Biology

rroberts@plos.org

on behalf of

Melissa Vazquez Hernandez, Ph.D.

Associate Editor

PLOS Biology

DATA POLICY:

Regardless of the method selected, please ensure that you provide the individual numerical values that underlie the summary data displayed in the following figure panels as they are essential for readers to assess your analysis and to reproduce it: Figs 1AB, 2, 3, 4BCD, 5B, 6, 7AB, 8ABCD, S1, S2, S3, S4, S5, S6, S7, S8, S9, S10, S11, S12, S13, S14, S16. NOTE: the numerical data provided should include all replicates AND the way in which the plotted mean and errors were derived (it should not present only the mean/average values).

CODE POLICY

DATA NOT SHOWN?

REVIEWERS' COMMENTS:

Reviewer #1:

In this revised version of the article "Viral and immune dynamics of human papillomavirus genital infections in young women", Tessandier and Elie et al. make appreciable updates to the study in response to the reviewers. Notably, the authors improved the statistical model by incorporating information about data censorship, removing the hypothesis of maximal effect duration, and including a sensitivity analysis related to the choice of priors. The authors also included an additional analysis of cell and cytokine associations broken down by infection status, identifying a new association with interferon gamma. Lastly, the authors include updates to the text regarding specific technical limitations and future directions such as CyTOF. That being said, the text is quite long and detailed. Some of the explanations of the model (or the validation of various aspects) could be moved into the methods text constrains become an issue. Overall, this exciting study has been improved by the authors' responses to the reviewers and this important work warrants publication. 

Reviewer #3:

The authors have addressed my comments very well.

---

## [Editor Report · Decision Letter 3]

22 Nov 2024

Dear Samuel,

Thank you for the submission of your revised Research Article "Viral and immune dynamics of genital human papillomavirus infections in young women with high temporal resolution" for publication in PLOS Biology. On behalf of my colleagues and the Academic Editor, Katherine Kedzierska, I am pleased to say that we can in principle accept your manuscript for publication, provided you address any remaining formatting and reporting issues. These will be detailed in an email you should receive within 2-3 business days from our colleagues in the journal operations team; no action is required from you until then. Please note that we will not be able to formally accept your manuscript and schedule it for publication until you have completed any requested changes.

IMPORTANT: Many thanks for providing the code to generate the figures in the CNSR depository. However, we are still missing that the figure legends mention the location of the code to generate the figure. Please put in each relevant figure legend the following: "The code to generate Figures X can be found in doi.org/10.57745/KJGOYZ". I have asked my colleagues to include this request alongside their own.

PRESS

Sincerely, 

Melissa

Melissa Vazquez Hernandez, Ph.D., Ph.D.

Associate Editor

PLOS Biology
